# Brain vascular stability relies on PAK2–cilia–PDGF-BB–HSPGs on basolateral side of endothelium

Shubhangi Prabhudesai[1,]*, Karthikeyan Thirugnanam[1,]*, Xuehong Song[5], Hua Yang[5], Mariella Errede[6], Francesco Girolamo[6], Thomas Neumann[8], Andrea Marzullo[7], Sepand Bafti[8], Kayla Vanderhoef[1], Kevin R Rarick[2], Andrew D Spearman[3], Amy Y Pan[4], Claudia Alvarez Alvarez[5], Jiyuan Yang[9], Fuming Zhang[9], Jonathan S Dordick[9], Daniela Virgintino[6], Lianchun Wang[5,]*, Ramani Ramchandran[1,]*

Endothelial cells (ECs) in the brain communicate with mural cells to facilitate vascular stability. Platelet-derived growth factor-BB (PDGF-BB)/platelet-derived growth factor receptor-$\beta$ (PDGFR-$\beta$) signaling mechanism at EC–mural cell interface helps stabilize the vasculature. How this paracrine signaling is mediated is not known. Our laboratory studies endothelial cilia, a microtubule-based organelle, and its role in promoting vascular stability. We discovered that brain endothelial cilia are located primarily on the basolateral side, and PDGF-BB is expressed in EC cilium. Thus, we hypothesized that endothelium cilium in conjunction with PDGF-BB on the basolateral side is responsible for mural cell recruitment. In this study, using a combination of zebrafish, mice, and human brain model systems, we have established a signaling paradigm wherein p21-activated kinase (PAK2) and ADP-ribosylation factor-13b (ARL13b) in ECs induce secretion of PDGF-BB. PDGF-BB associates with heparan sulfate proteoglycans (HSPGs) to form a gradient around ECs. Disrupting PAK2 affects ciliogenesis, HSPGs, and PDGF-BB gradient. We unravel a new mechanism involving endothelial cilia/PAK2-mediated PDGF-BB secretion, and retention by periendothelial HSPGs to promote vascular stability via recruiting mural cells.

## Introduction

During embryonic brain development, the vascular system in the brain develops in three phases: vasculogenesis, angiogenesis, and barriergenesis (Farrell & Risau, 1994; Engelhardt, 2003).

Primitive embryonic vessels cannot withstand shear stress, a tangential blood flow–related force on blood vessels. Supporting cells such as mural cells are needed to promote and maintain vascular stability. Vascular stability is established by paracrine signaling mechanisms involving brain endothelial cells (ECs) and mural cells such as pericytes. One of the well-established signaling paradigms in the field is the secretion of platelet-derived growth factor-BB (PDGF-BB) by ECs and its binding to platelet-derived growth factor receptor-$\beta$ (PDGFR-$\beta$) on pericytes (Betsholtz, 2004). EC-derived PDGF-BB has been suggested to bind to perivascular heparan sulfate (HS) (Abramsson et al, 2007) to confer pericyte recruitment and stabilization of vessels (Stratman et al, 2010; Stratman & Davis, 2012). Deficiency in HS (Abramsson et al, 2007) or deficient PDGF-BB binding to HS (Lindblom et al, 2003; Stenzel et al, 2009) impairs vascular development. Furthermore, HS in mural cells is dispensable for PDGF-BB signaling and pericyte recruitment in brain vasculature (Stenzel et al, 2009). These studies collectively suggest the importance of EC-derived PDGF-BB-HSPG interaction in facilitating vascular stability during embryonic development.

Previous work from our laboratory showed that PDGF-BB can induce primary cilium, a microtubule structure on brain microvascular ECs (Thirugnanam et al, 2022). Cilia are present on virtually all vertebrate cells. In ECs lining the vasculature, cilia are thought to exist on the apical surface of the ECs facing the lumen of blood vessels, and thus sense blood flow and serve as mechanosensors in multiple model systems (Nauli et al, 2008; AbouAlaiwi et al, 2009; Mohieldin et al, 2016; Pala et al, 2018). Within the vascular system, primary cilia are present on arteries, veins, lymphatics (Paulson et al, 2021), corneal endothelium, and endocardial and smooth muscle cells of arterial and airway endothelia (Luu et al, 2018).

[1]Division of Neonatology, Developmental Vascular Biology Program, Department of Pediatrics, Children's Research Institute (CRI), Medical College of Wisconsin, Milwaukee, WI, USA    [2]Division of Critical Care, Developmental Vascular Biology Program, Department of Pediatrics, Children's Research Institute (CRI), Medical College of Wisconsin, Milwaukee, WI, USA    [3]Division of Pediatric Cardiology, Developmental Vascular Biology Program, Department of Pediatrics, Children's Research Institute (CRI), Medical College of Wisconsin, Milwaukee, WI, USA    [4]Division of Quantitative Health Sciences, Department of Pediatrics, Children's Research Institute (CRI), Medical College of Wisconsin, Milwaukee, WI, USA    [5]Department of Molecular Pharmacology and Physiology, Byrd Alzheimer's Research Institute, University of South Florida, Tampa, FL, USA    [6]Bari University School of Medicine, Bari, Italy    [7]Department of Precision and Regenerative Medicine and Ionian Area, University of Bari, Bari, Italy    [8]Nortis Inc. Company, Seattle, WA, USA    [9]Departments of Chemistry and Chemical Biology, Center for Biotechnology and Interdisciplinary Studies, Department of Biological Sciences, Rensselaer Polytechnic Institute, Troy, NY, USA

Correspondence: lianchunw@usf.edu; rramchan@mcw.edu
*Shubhangi Prabhudesai, Karthikeyan Thirugnanam, Lianchun Wang, and Ramani Ramchandran contributed equally to this work

Endothelial cilia have been shown to be essential for developmental vascular integrity in zebrafish (Kallakuri et al, 2015; Eisa-Beygi et al, 2018). Previously, we showed that the vascular stability *redhead* (*rhd*) *p21-activated kinase 2a* (*pak2a*) mutant with cerebral hemorrhages was rescued by cilium (*Arl13b* mRNA) expression (Thirugnanam et al, 2022), which implies that ciliogenesis mechanisms mediated by PAK2-ARL13b signaling is responsible for vascular stability. However, the underlying endothelial cilium-based mechanisms associated with brain vascular stability were not known, which were investigated here. EC cilia are known to control endothelial permeability by regulating the expression and localization of junction proteins (Diagbouga et al, 2022). Apically localized cilia in zebrafish ECs have been suggested to regulate shear stress, Notch activation, and *foxc1b* expression, which leads to recruitment of vascular mural cells to facilitate vascular stabilization (Chen et al, 2017). Current thinking for endothelial cilia in promoting vascular stability in developing endothelium is that apical EC cilia, through shear stress mechanism, facilitate signaling molecules that promote interaction with mural cells that are located basolaterally. Based on current findings, we postulate a different model wherein brain endothelial cilia located on the "abluminal (basolateral)" side are responsible for facilitating interactions with mural cells to promote vascular stability. We investigate this hypothesis in this study and identified the underlying mechanisms associated with basolateral EC cilia–mediated embryonic vascular stability.

# Results

### Pak2a rhd mutants show defective brain EC junctions, and pericyte recruitment to vascular cilia

Brain vascular instability is associated with disruption of endothelial cell junctions and defective pericyte recruitment (Gaengel et al, 2009). We investigated vessel junctions in *pak2a* mutants by staining for claudin 5, an established marker for brain EC junctions (Fig 1) (Morita et al, 1999). Cell junctions were assessed in *pak2a*^mi149/mi149^ homozygous (hom) bleeders (Fig 1A, A', and A''), *pak2a*^mi149/+^ heterozygous (het) non-bleeders (Fig 1B, B', and B''), and *pak2a*^+/+^ WT (Fig 1C, C', and C''), in the primordial hindbrain channel (PHBC) 80 microns posterior of the middle cerebral vein (MCeV) at the 52-h post-fertilization (hpf) time point. Compared with *pak2a* WT (Fig 1C'') and *pak2a* het non-bleeders (Fig 1B''), *pak2a* hom bleeders (Fig 1A'') showed loss of EC junctions (Fig 1D). We also investigated *pdgfrb*⁺ pericytes and cilia in *pak2a* WT, *pak2a* het non-bleeders, and *pak2a* hom bleeders at 52 hpf. To assess pericytes and cilium coverage on ECs, triple-transgenic line (*pdgfrb*: GFP; *kdrl*: mCherry; and *beta-actin*: Arl13b-GFP) was created, which was crossed into *pak2a* hets. Incross of these hets resulted in triple-transgenic *pak2a* hom embryos. *Pdgfrb* positivity is marked as star-shaped pericytes (Fig 2A'–C' inset, white and red asterisks), and *arl13b* positivity is marked as string-shaped cilia (Fig 2B', white arrow) in the hindbrain vasculature. Representative images for *pak2a* WT (Fig 2A and A'), *pak2a* het non-bleeders (Fig 2B and B'), and *pak2a* hom bleeders (Fig 2C and C') at 52 hpf are shown. Quantifications of pericytes and cilia are described in the Materials

and Methods section. We compared the number of pericytes (Fig 2D), distance of pericytes from EC cilia (Fig 2E), and vessels (Fig 2F), and percentage of pericytes close to vessels with (Fig 2G) or without (Fig 2H) cilia. Comparisons were made between *pak2a* hom bleeder, *pak2a* het non-bleeder, and *pak2a* WT embryos. *pak2a* hom bleeder and *pak2a* het non-bleeder showed fewer pericytes (Fig 2D). There was no significant difference in the distance of pericytes from EC cilia (Fig 2E) or vessels (Fig 2F) in all three groups. Compared with WT brain where most of the pericytes are close to vessels with cilia (Fig 2G), in *pak2a* hom bleeders, a median of 25% are next to vessels with cilia ($P = 0.04$) and 75% are next to vessels without cilia ($P = 0.04$) (Fig 2H). The distribution is opposite (67% vessel with cilia and 33% vessels without cilia) for heterozygous non-bleeders (Fig 2G and H). Finally, we noticed *pdgfrb*⁺ pericytes with varying arm lengths (short arm <5 $\mu$m, white asterisk, Fig 2A'–C'; long arm >5 $\mu$m, red asterisk, Fig 2B' and C') in our transgenic fish (Fig S1). *pdgfrb*⁺ pericytes with short-arm and long-arm lengths showed equal distribution in *pak2a* hom bleeders (Fig S1). This result contrasts with *pak2a* het non-bleeder and WT data where pericytes with short-arm length were greater in number than pericytes with long-arm length (Fig S1). These results collectively suggest that *pak2a* regulates EC junctions, pericyte morphology, and proximity to EC cilia.

### Brain endothelial cilia are localized at the basolateral surface

For pericytes in tissue to interact with endothelial vascular cilia, we hypothesized that the brain endothelial cilia on the basolateral side would greatly facilitate the interaction. Basolateral endothelial cilium localization in lymphatic vessels supports this hypothesis (Paulson et al, 2021). We investigated brain EC cilium location in multiple model systems including human primary microvascular brain ECs embedded in a collagen matrix under laminar shear stress, a zebrafish model, mice, and human brains. In these models, brain EC cilia were stained either for Arl13b ciliary marker or for acetylated $\alpha$-tubulin ciliary marker, or for both. Brain microvascular ECs in microfluidic chips subjected to steady-state flow formed a tube-like configuration (Fig 3A), which distinguished the lumen and the basolateral side of the endothelium. Our data show that cilia were located predominantly in the basolateral side (Fig 3A and B, arrowheads) of the endothelium. In zebrafish, we used a double-transgenic line that expresses mCherry in the vasculature and fusion ciliary protein marker Arl13b-GFP under *b-actin* promoter to investigate brain EC cilium location (Fig 3C and C'). Remarkably, EC cilia were found jutting out into the abluminal (basolateral) side (Fig 3C', white arrows). Quantification (Fig 3D) shows that in MTA and MCeVs in zebrafish, the percentage of cilium frequency is roughly equal across the luminal and abluminal side. In mouse cortex sections, we performed immunohistochemistry for Arl13b protein (Fig 3E, black arrow). Quantification shows that abluminal cilium length is ~4–5 $\mu$m, whereas luminal cilium length is 3–4 $\mu$m (Fig 3E'). Furthermore, more than double the number of EC cilia are localized on the basolateral abluminal side compared with apical luminal side (Fig 3E''). Finally, in a 22-wk fetal human brain cortex, we observed ARL13B cilia in abluminal location on a CD31 vessel (Fig 3F, white arrows). The CD31 microvessels ranged from capillaries, small arterioles, and venules, and ARL13B⁺ cilia

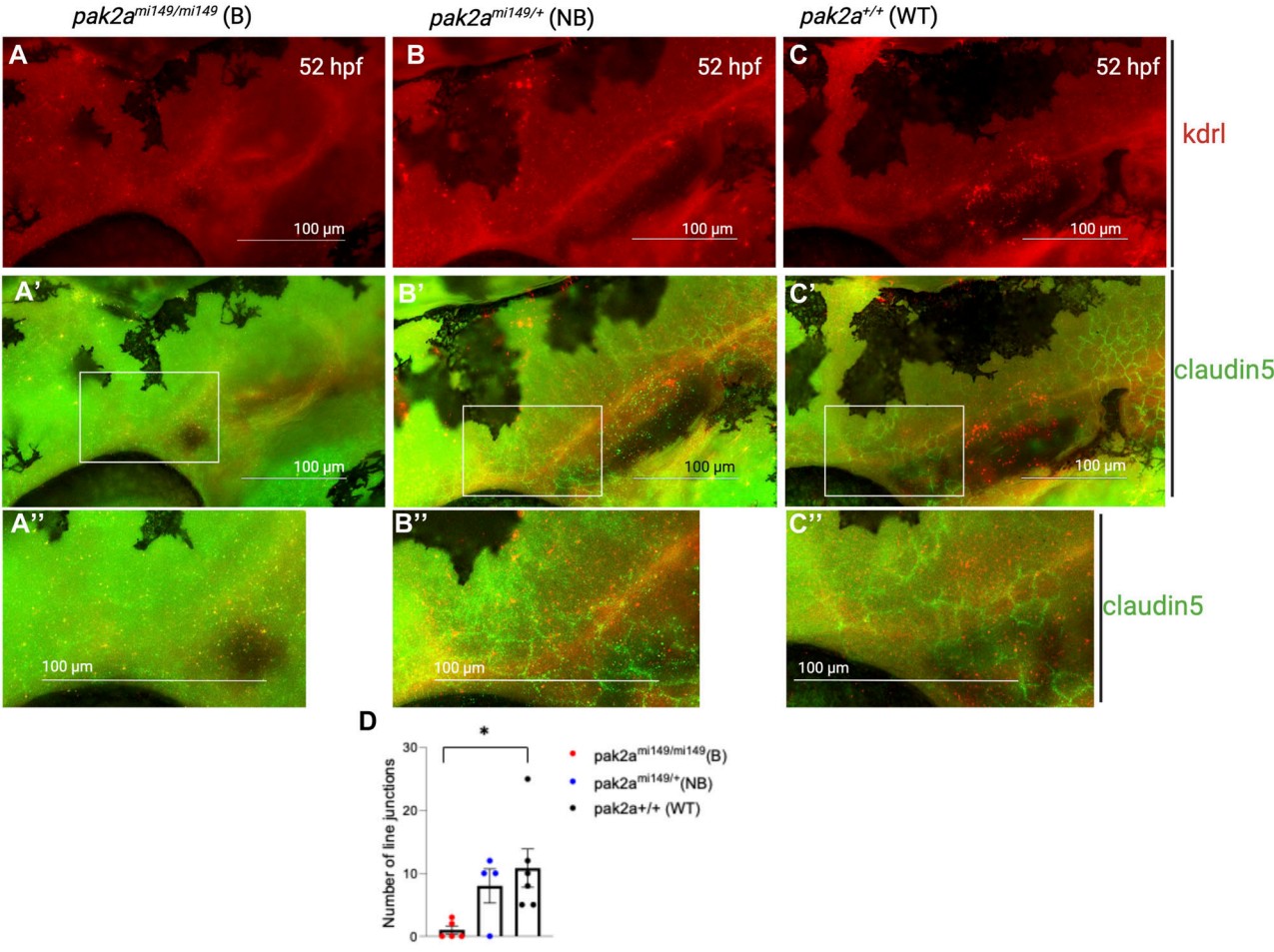

**Figure 1. *rhd* mutant and endothelial cell junction analysis.**
**(A, B, C)** *pak2a rhd$^{mi149/mi149}$* (homozygous) bleeder (B), *pak2a rhd$^{mi149/+}$* (heterozygous) non-bleeder (NB), and *pak2a$^{+/+}$* WT 52-h post-fertilization (hpf) zebrafish embryos, respectively, stained for kdrl (VEGFR-2) antibody (red). PHBC is the primordial hindbrain channel. **(A', B', C')** Immunostained for claudin 5 junctions using claudin 5 antibody (green) at 52 hpf. **(A", B", C")** White rectangle box is magnified in (A", B", C"), respectively. **(D)** Quantification of the line junctions in 80 μm length of PHBC. *pak2a$^{+/+}$* WT (n = 6), *pak2a rhd$^{mi149/+}$* (heterozygous) non-bleeder (NB) (n = 4), and *pak2a rhd$^{mi149/mi149}$* (homozygous) bleeder (B) (n = 5). Scale bars are 100 μm. A *t* test was performed to compare junction differences between the groups, and *P < 0.05 was considered statistically significant.

were measured in three distinct zones of telencephalon wall, namely, ventricular zone (VZ), subventricular zone (SVZ), and cortical plate (CP). Quantification shows a propensity for brain EC cilia to be present in the basolateral side versus apical side in the VZ and SVZ vasculature (Fig 3F'). In the VZ and SVZ, ARL13B$^+$ cilia were detected only on the surface of CD31$^+$ ECs of microvessels no larger than 15 μm in diameter, with their average internal diameters between 7.45 μm in VZ and 7.73 μm in SVZ, respectively. In the CP, ARL13B$^+$ cilia were not detected on the surface of CD31$^+$ ECs of microvessels. The density of ARL13B$^+$ cilia in VZ and SVZ vasculature was similar (*P* > 0.05). On average, in the VZ there were 0.57 ± 0.6 abluminal cilia/100 μm–cumulative vessel length (CVL) (1 cilium on 175 μm of explored microvessels) and 0.24 ± 0.42 luminal cilia/ 100 μm–CVL (1 cilium on 416 μm). In the SVZ, there were 0.67 ± 0.9 abluminal cilia/100 μm–CVL (1 cilium on 149 μm) and 0.27 ± 0.41 luminal cilia/100 μm–CVL (1 cilium on 370 μm) (Fig 3F'). These data collectively in multiple model systems suggest that brain endothelial cilia are primarily located on the basolateral side,

which can facilitate interactions with cells of the brain microenvironment such as mural cells.

**PDGF-BB secretion in brain ECs is induced by *Pak2a* and cilium signals**

Past work has provided extensive evidence that the platelet-derived growth factor-BB (PDGF-BB) secreted by ECs binds to PDGFR-β on pericytes to promote vascular stability. This concept is based on mouse knockouts for PDGF-BB or PDGFR-β that die perinatally and show reduced mural cell coverage of endothelium, increased pericyte/mural cell detachment, capillary dilation, and widespread microaneurysms resulting in severe hemorrhaging (Lindahl et al, 1997; Lindblom et al, 2003; Betsholtz, 2004; Abramsson et al, 2007; Gaengel et al, 2009). Our previous work also showed that PDGF-BB was a potent inducer of PAK2a and ARL13b signals in brain microvascular ECs (Thirugnanam et al, 2022). Furthermore, *pak2a* hom bleeders showed fewer *pdgfrb$^+$*

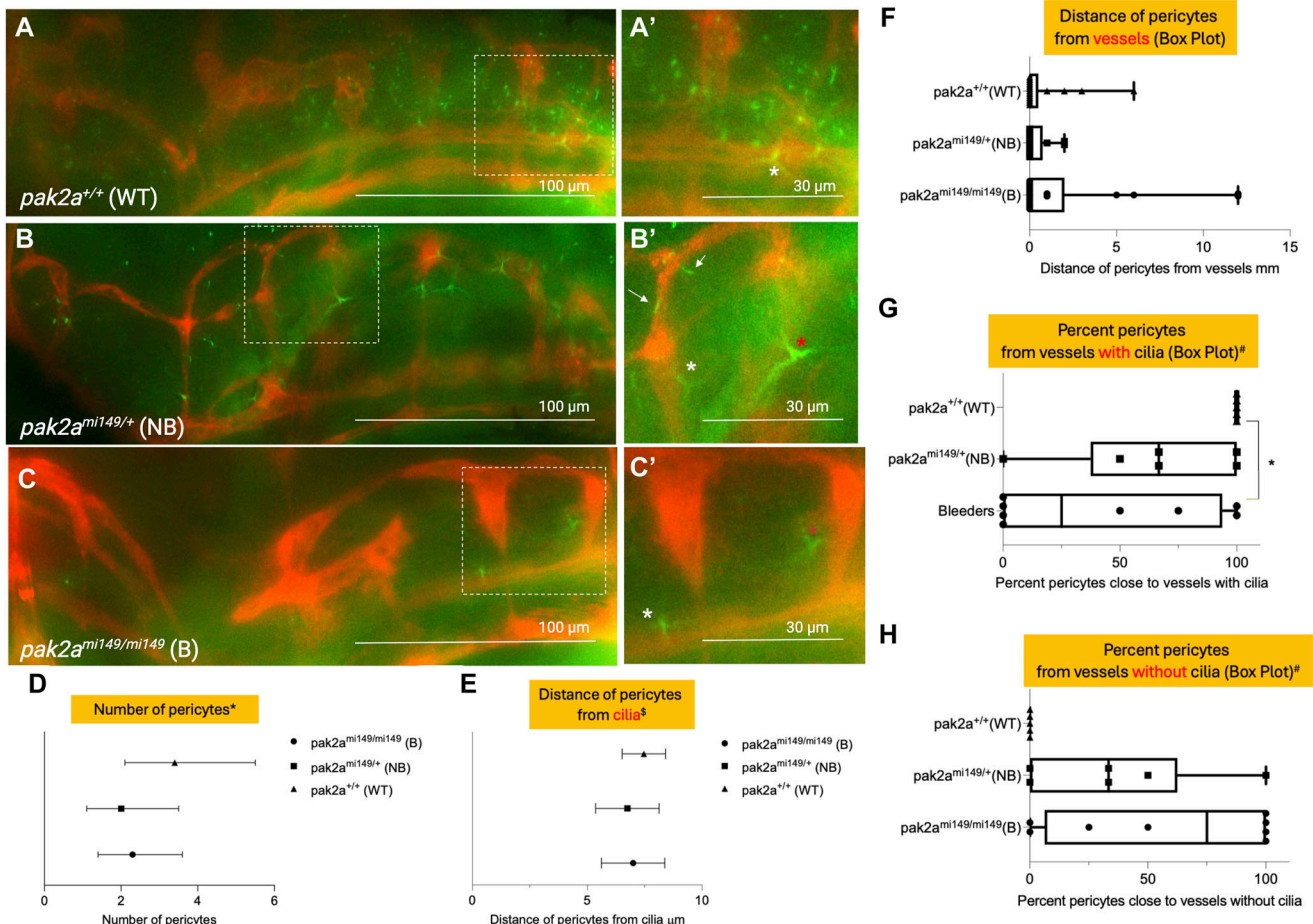

**Figure 2. *rhd* mutant and pericyte analysis.**
**(A, B, C)** *pak2a+/+* WT, *pak2a rhd mi149/+* (heterozygous) non-bleeder (NB), and *pak2a rhd mi149/mi149* (homozygous) bleeder (B), respectively, at 52-h post-fertilization (hpf) transgenic zebrafish embryos derived from an incross of *pak2a rhd mi149/+* heterozygous in the background of zebrafish line Tg(*pdgfrb*: GFP; *kdrl*: mCherry; and *beta-actin*: Arl13b-GFP). **(A', B', C')** Magnified regions of the white dotted boxes in the respective panels. **(A', B', C')** Star-shaped cells ((B', C'), red asterisk) are long-arm pericytes and short-arm pericytes ((A', B', C'), white asterisk), and white arrows (B') are cilia. **(D)** Number of *pdgfrb+* pericytes quantified between *rhd* WT, heterozygous (mi149/WT), and homozygous (mi149/mi149) embryos. **(A', B', C')** Scale bars in (A, B, C) are 100 *μm*, and in (A', B', C') are 30 *μm*. A rectangular area (length = 190 *μm* posterior to the MCeV and width = 150 *μm* dorsal to the PHBC) was analyzed in all images. **(E)** Distance of pericyte from endothelial cilia. **(F, G, H)** Distance of pericytes from vessels (F), and pericyte distribution from vessels with (G) and without (H) cilia are plotted across the three sample groups. **(F, G, H)** Panels are box plots. *P < 0.05. n = 8 *pak2a* homozygous bleeders (B) (mi149/mi149), n = 6 *pak2a* heterozygous non-bleeders (NB) (mi149/WT), and n = 5 *pak2a* WT (WT/WT). Panel (F) shows estimated means and 95% confidence intervals (CIs). Panel (G) shows means and standard error of means. Results were not statistically significant. **(D, E)** GLM with the Poisson distribution or ANOVA with Dunnett's test for multiple comparisons adjustment was used for statistical analysis of panels (D, E), respectively, and results were not statistically significant. **(E, F)** Adjusted *P* = 0.37 (WT versus B) and *P* = 0.27 (WT versus NB); (F) and *P* = 0.95 (WT versus B) and *P* = 0.88 (WT versus NB) (E). Panel (F) shows all data points and box plots. The Kruskal–Wallis test was used. *P* = 0.68. For panels (G, H), all data points and box plots were plotted. **(G, H)** Kruskal–Wallis test along with Dwass–Steel–Critchlow–Fligner method for multiple comparisons adjustment (G, H) was used for statistical analysis. Bleeders versus WT groups were compared and *P* = 0.04 (G) and *P* = 0.04 (H).

mural cells, and of the ones present, most mural cells misaligned to brain vessels with no cilia (Fig 2). Thus, we hypothesized that Pak2a may induce PDGF-BB signal to promote mural cell recruitment. We tested this hypothesis in primary human brain microvascular ECs and investigated PDGF-BB levels by ELISA in supernatants from *PAK2* and *ARL13b* knockdown brain ECs (Fig 4A and B). Indeed, *PAK2* and *ARL13b* knockdown human brain ECs both show reduced PDGF-BB secretion in the supernatant (Fig 4A and B) with *PAK2* knockdown showing more pronounced effect (50% decrease) than *ARL13b* knockdown (20% decrease) EC supernatants compared with control ECs.

## PDGF-BB ligand is expressed in a gradient fashion proximal to brain ECs in vertebrates

PDGF-BB contains a retention motif, an HS binding sequence that is necessary for interaction with basement membrane HSPG molecules localized on the basolateral side of the endothelium (Lindblom et al, 2003). This retention is necessary for pericyte recruitment and has led to the hypothesis that PDGF-BB released from ECs is retained by the EC basement membrane to promote a gradient around these cells. However, to our knowledge, this gradient has never been observed in vivo in the literature. Here, we

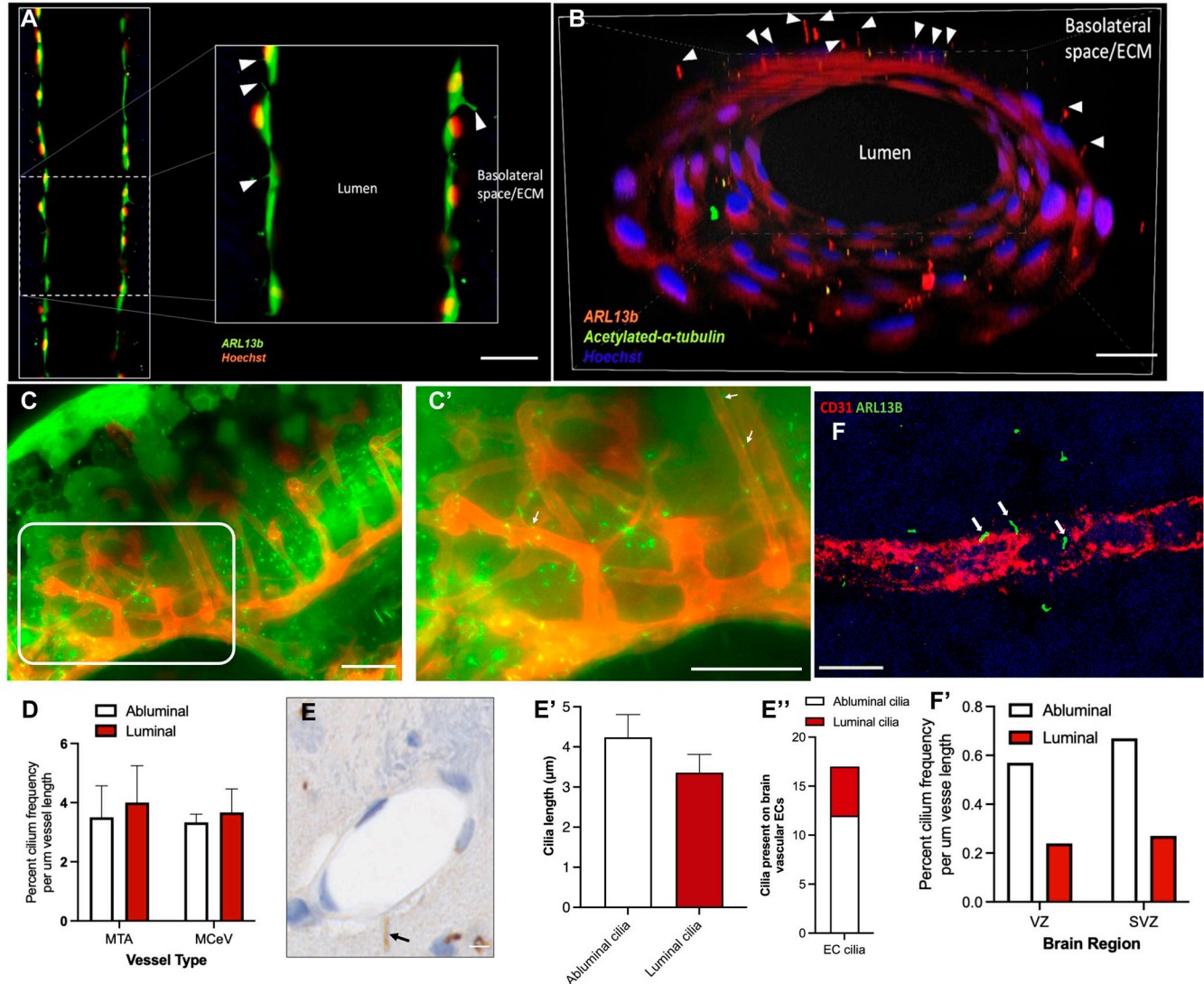

**Figure 3. Basolateral EC cilium location in multiple species.**
**(A)** Human microvascular brain ECs cultured in a microfluidic chamber and stained for ciliary marker ARL13b and acetylated $\alpha$-tubulin in the cross-sectional plane; the white dotted box was enlarged in the middle. **(B)** Transverse plane of the vessel tube. White arrowheads indicate ARL13b cilia projecting into the basolateral space. Scale bars: 100 $\mu m$. **(C, C')** Fluorescent image of a 52-hpf transgenic zebrafish Tg(*flk*:mCherry; *b-actin*:Arl13b-GFP) expressing mCherry in the vasculature and mouse Arl13b-GFP fusion protein in the cilia. **(C')** White boxed area is enlarged in (C'). White arrows in (C') point to basolateral zebrafish endothelial cilium location. The scale bar in (C, C') is 50 $\mu m$. **(D)** Quantification of zebrafish endothelial cilia (luminal versus abluminal location) in metencephalic artery (MTA) and middle cerebral vein (MCeV). N = 4 for each vessel group was quantified for the presence of abluminal versus luminal cilia. An unpaired *t* test showed no difference in the cilium location between the two vessel groups. **(E)** Immunohistochemistry for abluminal cilia from representative cortical microvessels from *ARL13b-EGFP* transgenic adult mice (*C57BL/6* background) showing luminal versus abluminal cilium localization stained with anti-GFP. The black arrow points to Arl13b-positive abluminal cilia. Scale bars: 100 $\mu m$. **(E')** Quantification of Arl13b cilium length on the abluminal versus luminal side of mouse endothelial vessels. **(E")** Frequency of cilia on the abluminal versus luminal side of mouse endothelial vessels. A total of 30 endothelial vessel-like structures were quantified. **(F)** Immunofluorescence staining of a 22-wk-old human brain for ARL13b protein (green) and CD31 vessel staining (red). White arrows point to cilia in the basolateral side of the vessel. Scale bars: 10 $\mu m$. **(F')** Quantification of endothelial cilial frequency in the ventricular zone (VZ) and subventricular zone (SVZ) in human brain sections. Scale bar in (F) is 10 $\mu m$.

stained for endomucin (vessel marker) (Fig 4C) and PDGF-BB (Fig 4D) in WT *Ext1$^{fl/fl}$* mouse E11.5 cortex and clearly observed a gradient of PDGF-BB surrounding the endothelial vasculature (Fig 4E). We also stained PDGF-BB protein in zebrafish brain, which also showed pockets of robust expression proximal to vessels (Fig 4F, arrows), and a gradient-like expression pattern (Fig 4F' and F", orange boxes). To evaluate whether brain ECs in culture retain

PDGF-BB on the HS chains, we treated human brain microvascular ECs with heparinase (0.2 IU/ml) for 2 h. Heparinase cleaves HSPGs from cell surface (Shaya D et al, 2010). Indeed, PDGF-BB levels are higher in heparinase-treated control ECs in two independent experiments (Fig 4A and B, heparinase-treated samples, $P <$ 0.005 or $P <$ 0.05 compared with the untreated control). These results collectively suggest that PDGF-BB is found immediately

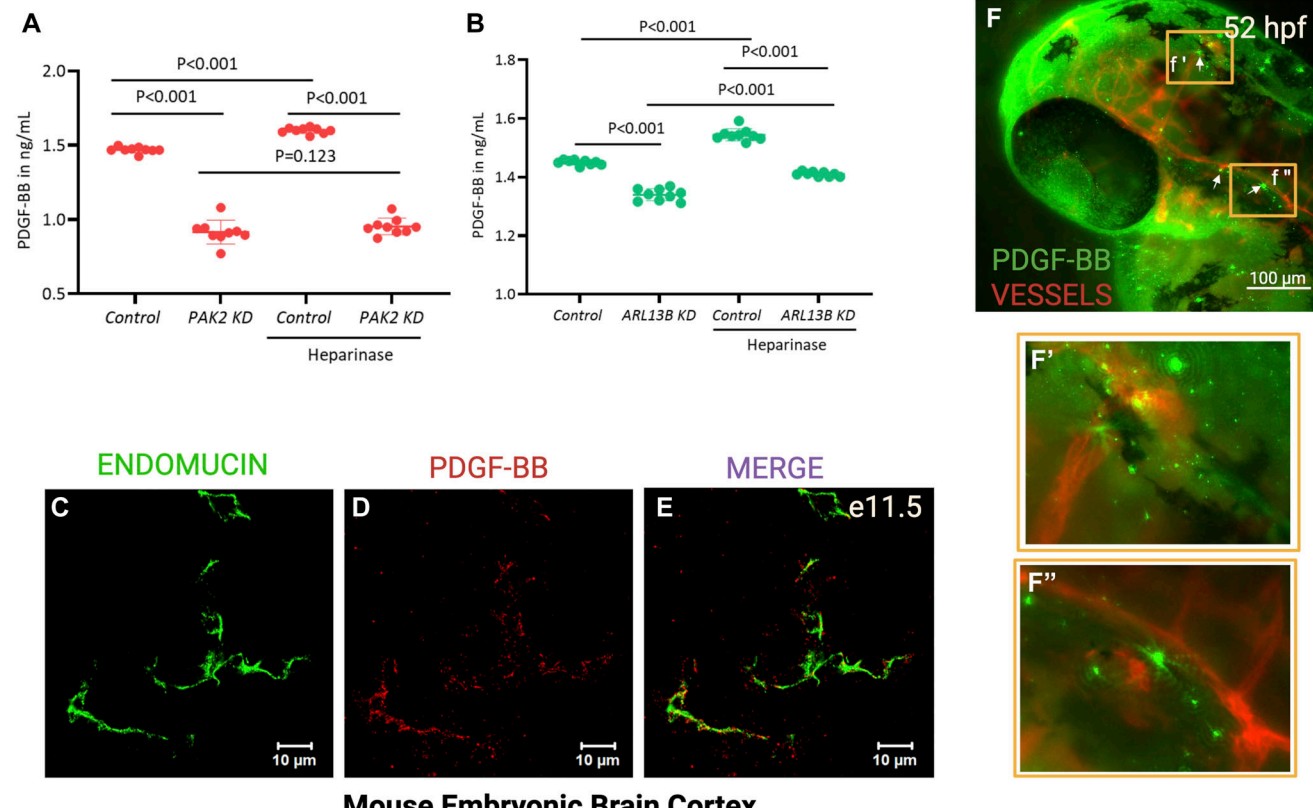

**Figure 4. PDGF-BB expression in brain ECs in vitro and in vivo.**
**(A, B)** ELISAs performed on supernatants from control siRNA, *PAK2* siRNA (A), and *ARL3b* siRNA (B) brain ECs with or without heparinase II treatment. **(C, D, E)** Immunofluorescent brain images of e11.5 WT *Ext1^fl/fl* mouse cortex stained for endomucin (green), PDGF-BB (red), and merged image. **(F)** Immunofluorescence image of PDGF-BB antibody stained (green) in a Tg(*flk*:mCherry) (red) 52-hpf zebrafish embryo. **(F, F', F")** White arrows point to PHBC. **(F, F', F")** Orange box in panel (F) is magnified in (F', F") showing PDGF-BB gradient-like expression close to red blood vessels in the brain. The scale bar in (F) is 100 *μm*.

outside of vertebrate brain vessels and is retained on the EC surface and surrounding extracellular matrix via binding to HSPG to create a gradient-like pattern.

### PDGF-BB gradient is disrupted in *pak2*- and heparan sulfate–deficient background in vivo

As PAK2 induces PDGF-BB levels in brain ECs in vitro (Fig 4A), we investigated whether the number of PDGF-BB sources and gradients was different between embryonic brains of *pak2a* WT fish (Fig 5A and A') and *pak2a* hom bleeder fish (Fig 5B). Qualitatively, PDGF-BB staining is vastly reduced in *pak2a* hom bleeders (Fig 5B) compared with *pak2a* WT fish (Fig 5A). In the head (Fig 5C and D) and PHBC (Fig S2A and B), the number of PDGF-BB sources remains unchanged between *pak2a* hom bleeders compared with *pak2a* WT embryos (Fig 5C), but in both cases, the number of PDGF-BB gradients is vastly decreased in *pak2a* hom bleeders compared with *pak2a* WT embryos (Figs 5D and S2B). We further evaluated the PDGF-BB gradients by measuring the number of PDGF-BB rings of expression within a gradient (Fig 5E) and the distance of the farthest point in the gradient from the initiating source (Fig 5F). The number of rings was statistically significant (**$P < 0.01$, Fig 5E), whereas the distance of furthest ring from the source was not

(Fig 5F). In heart (Fig S2C and D), both PDGF-BB sources and PDGF-BB gradients are decreased in *pak2a* hom bleeders compared with *pak2a* WT embryos. Others have hypothesized that the PDGF-BB gradient is formed because of interactions with HSPGs secreted from ECs (Betsholtz, 2004). The abundance and structure of HS in each cell type are determined by the cellular expression of the enzymes responsible for HS biosynthesis and remodeling (Bishop et al, 2007). Thus, we investigated the status of the PDGF-BB gradient surrounding endomucin-positive vasculature in e11.5 embryos with endothelial-specific (*TIE2-CRE*) knockout of the HS copolymerase enzyme *Exostosin-1* (*Ext1*) (*Ext1^ECKO*), in which endothelial HS expression is abolished (Wang et al, 2005; Qiu et al, 2018). Indeed, we observed that the PDGF-BB gradient is disrupted upon knockout of EC-HS expression (*Ext1^ECKO*) (Fig 5K) compared with littermate-matched E11.5 control (*Ext1^fl/fl*) (Fig 5H) mouse brains. We quantified the PDGF-BB gradient in the mouse brain and found that the PDGF-BB gradient extended further from endomucin-positive vessels in the *Ext1^fl/fl* brain section (Fig 5I and I') compared with a diminished gradient in the *Ext1^ECKO* brain section (Fig 5L and L'). These data collectively suggest that *pak2a* and HS biosynthesis is intricately linked on the basolateral side of the endothelium to promote PDGF-BB gradient formation.

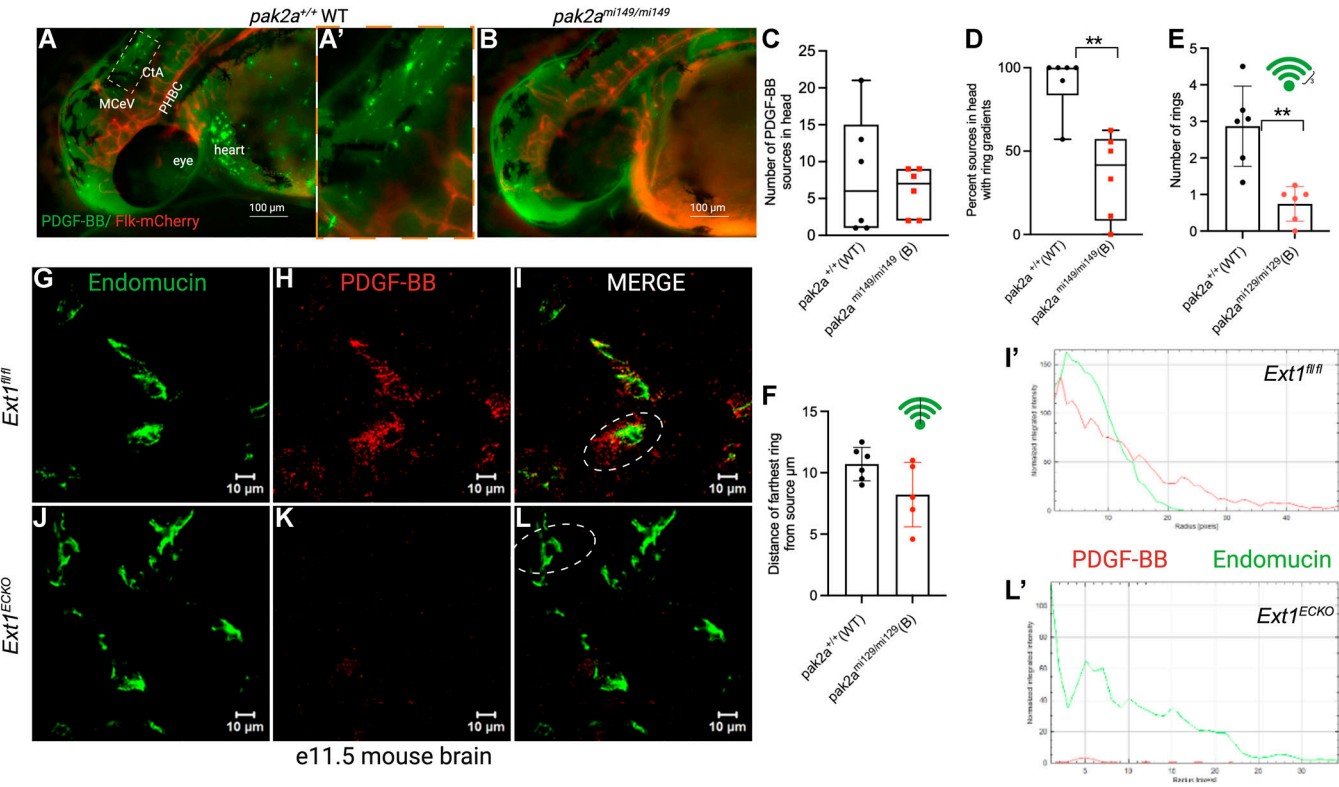

**Figure 5. PDGF-BB gradient changes in *pak2a* zebrafish brain and *ext1* endothelial knockout mouse brain.**
**(A, A′, B)** Immunofluorescence images of PDGF-BB antibody stained (green) in a Tg(*flk*:mCherry) (red) 52 hpf *pak2a*^+/+ WT (WT) (A, A′) and *pak2a rhd*^mi149/mi149^
homozygous (B) zebrafish embryo. MCeV is mid-cerebral vein, PHBC is primordial hindbrain channel, and CtA stands for central arteries. **(A, A′)** Dotted yellow box in
panel (A) is magnified in (A′). **(C, D)** Quantification of the number (C) and percentage of PDGF-BB ring gradients (D) in *pak2a* WT (+/+) and *pak2a* homozygous bleeders (B)
(mi149/mi149). N = 6 for each group. The Mann–Whitney–Wilcoxon test was used to analyze difference in percent sources in the head with ring gradients. **P = 0.0072.
**(E, F)** Number of PDGF-BB rings and (F) distance of the farthest PDGF-BB ring from the center of gradient. **(E, F)** Green Wi-Fi–like symbols ((E), three rings; and (F), black
line used to measure the distance) are included to indicate PDGF-BB gradient in fish. For (E) (number of PDGF-BB rings), PDGF-BB antibody–labeled heads (n = 6) were
quantified for *pak2a*^+/+ WT and *pak2a rhd*^mi149/mi149^ bleeders. Unpaired Welch's *t* test was used to analyze the differences in the number of rings, **P = 0.0036. For (F)
(farthest ring distance), *pak2a*^+/+ WT (n = 6) and *pak2a rhd*^mi149/mi149^ bleeders (n = 5) were used. One of the *pak2a rhd*^mi149/mi149^ bleeders had no ring gradients around any
of the PDGF-BB sources and thus was not used for quantification. Unpaired Welch's *t* test was used to analyze the differences in farthest ring distances, which was
statistically insignificant, P = 0.1056. **(G, H, I, J, K, L)** Immunofluorescent brain images of e11.5 mouse cortex stained for endomucin (green) and PDGF-BB (red) in
littermate controls *Ext1*^fl/fl^ (*Ext1*^fl/fl^) (G, H, I) versus endothelial-specific *Ti2Cre*^+^*Ext1*^fl/fl^ knockout (*Ext1*^ECKO^) embryos (J, K, L). **(I′, L′)** Quantification of the PDGF-BB and
endomucin signals in *Ext1*^fl/fl^ control (I′) and *Ext1*^ECKO^ (L′) mouse brains using the Fiji Radial Profile Extended plugin to calculate spatial gradients around endothelial
cells. Compared with *Ext1*^fl/fl^ control, *Ext1*^ECKO^ mouse brains exhibit a markedly attenuated PDGF-BB gradient. Scale bars in (A, B) are 100 μm. Scale bars in (G, H, I, J, K, L)
are 10 μm. All data points in (C) were plotted along with the median values. GLM with the Poisson distribution was used to examine the differences in the number of
PDGFB sources in the head. P = 0.1920.

## PDGF-BB is found in brain EC cilia in vitro

As HS and brain EC cilia are observed on the basolateral side, and
bound PDGF-BB is released from ECs upon HS cleavage (Fig 4A
and B), we investigated whether PDGF-BB is present inside the
brain EC cilium, as cilium is a source of ligands for signaling
(Malicki & Johnson, 2017; Pala et al, 2017). We performed im-
munofluorescence for ARL13B and PDGF-BB proteins in brain ECs
(Fig 6A and A′). Indeed, we observed PDGF-BB enriched in the
brain EC cilia (Fig 6A′, box magnified). Previously, we had
reported that some brain ECs in the G0 cell cycle phase
exhibit 2 ciliary phenotypes (Thirugnanam et al, 2023). In the
immunofluorescence image, note two ciliary dot-like structures
in brain ECs (Fig 6A′). Interestingly, PDGF-BB was found to be
colocalized to a single cilium (Fig 6A′, arrow). The significance of

the PDGF-BB cilium localization in one of the two cilia is not
known. Because brain EC cilia are tiny (1–2 μm), they are often
observed as dot-like structures and are difficult to visualize the
conventional axoneme–basal body structure of the cilium. In the
past, upon treatment of brain ECs with PDGF-BB, we have ob-
served the entire ciliary structure: basal body (centrin2) and
axoneme (ARL13b) in primary human brain ECs (Thirugnanam
et al, 2023). Furthermore, we showed that PDGF-BB induced
PAK2 and ARL13b protein levels in brain ECs (Thirugnanam et al,
2022). These results collectively suggest a feedforward and
feedback signaling loop exists wherein PDGF-BB promotes
EC–ciliogenesis (PAK2a-ARL13b) signaling axes, which in turn
promotes more PDGF-BB secretion from ECs. To test the
feedforward–feedback mechanism in brain ECs, we compared ARL13b
and PAK2 expression in lysates from *control* siRNA and efficacy-

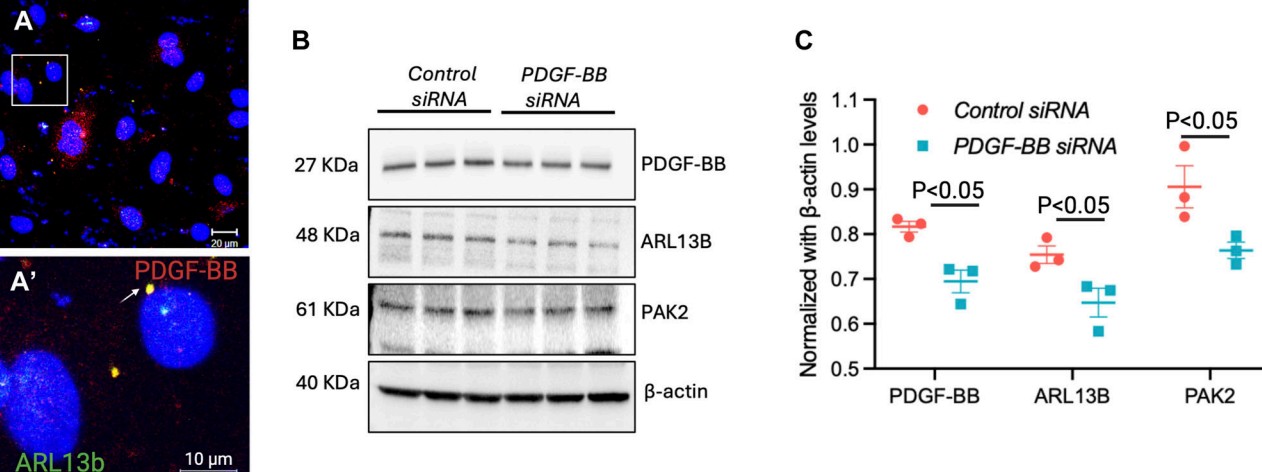

**Figure 6. PDGF-BB in brain endothelial cilia and its regulation of ARL13b and PAK2 signals in brain endothelial cells.**
**(A, A′)** IF for cilia ARL13b (green) and PDGF-BB (red) in brain microvascular ECs. **(A, A′)** Boxed region in panel (A) is zoomed in panel (A′). An arrow indicates a brain EC cilium that is colocalized with PDGF-BB. **(A, A′)** Scale bars: 20 μm for (A) and 10 μm for (A′). **(B)** Western blot of brain endothelial cell lysates from cells transfected with *control* or *PDGF-BB* siRNA. PDGF-BB, ARL13B, PAK2, and β-actin proteins were probed with specific antibodies and blots quantified by ImageJ. **(B, C)** Quantification data of the Western blots in panel (B). *n* = 3 in each experimental group. Results are presented as the mean ± SEM. Statistics were performed using paired *t* tests. *P* < 0.05 was considered significant.
Source data are available for this figure.

confirmed *PDGF-BB* siRNA brain ECs (Fig 6B). Indeed, we noticed a reduction in both ARL13B and PAK2 proteins in *PDGF-BB* siRNA ECs compared with *control* siRNA ECs (Fig 6C), confirming the feedforward–feedback PDGF-BB-PAK2/ARL13B-PDGF-BB regulatory mechanism in brain ECs.

## PAK2/ARL13b–heparan sulfate proteoglycan connection in vitro and in vivo

PAK2 and ARL13b induce PDGF-BB in brain ECs (Fig 4A), and PDGF-BB is retained by HS on the cell surface and within peri-endothelial cellular matrix (Fig 4D). Thus, we investigated the status of HS expression in *PAK2* and *ARL13b* loss-of-function ECs (Fig 7A and B). We performed qRT-PCR for genes of HS chain biosynthesis (*EXT1, EXT2*) and sulfation modification and remodeling (*NDST1, NDST2, HS6ST1, HS6ST2, HS6ST3, HS2ST1, SULF1, SULF2*) in *PAK2* siRNA (Fig 7A) and *ARL13b* siRNA (Fig 7B) brain ECs and compared their expression with control siRNA brain ECs (Fig 7A and B). Four of the enzymes showed up-regulation (*EXT1, HS6ST1, HS6ST2,* and *HS2ST1*) in *PAK2* siRNA brain ECs, five enzymes showed no change, and one enzyme showed down-regulation (*NDST1*) (Fig 7A). For *ARL13b* siRNA ECs, 2 enzymes showed up-regulation (*HS6ST2* and *HS6ST3*), five enzymes showed no change, and three enzymes showed down-regulation (*EXT2, NDST1,* and *NDST2*) (Fig 7B). *HS6ST2, HS6ST3,* and *NDST1* enzymes are regulated in both *PAK2* and *ARL13b* siRNA ECs compared with control ECs. qRT-PCR data in human brain ECs with *PAK2* or *ARL13b* siRNA suggest that HS chain biosynthesis and modifications are likely altered under PAK2 and/or ARL13b modulations. As we observed a greater number of HS enzyme changes in *PAK2* siRNA ECs, we isolated HS from *PAK2* siRNA brain ECs and performed disaccharide composition analysis, which showed a global reduction in sulfation, including N-sulfation (NS), 2-O-sulfation (2S),

and 6-O-sulfation (6S), in HSPGs on primary brain ECs (Fig 7C). Defective N-sulfation abolishes HS to bind PDGF-BB (Abramsson et al, 2007). We also performed PDGF-BB binding assay on *PAK2* siRNA ECs and *ARL13b* siRNA ECs using an ELISA that detects cell surface–bound PDGF-BB (Fig 7D). Heparinase I treatment of ECs was used as a positive control. Indeed, *PAK2* siRNA ECs and *ARL13b* siRNA ECs show reduced PDGF-BB binding to ECs compared with control siRNA ECs (Fig 7D). Finally, we stained for HS in *pak2a* WT fish (Fig 7E) and *pak2a* hom fish (Fig 7F) and found little to no staining on vessels prone to hemorrhage in *pak2a* hom fish. Collectively, these data suggest that PAK2 is required for appropriate HS biosynthesis and sulfation on brain EC surface to facilitate HSPG formation and PDGF-BB binding.

## *Pak2a*: EC cilia and role of HSPG

Thus far, we have provided evidence that PAK2 can induce PDGF-BB expression in brain ECs, and influences HSPG modifications on the surface of brain ECs, which is necessary for PDGF-BB binding to cell surface. The EC cilium status under *pak2* loss-of-function conditions remains unknown. To investigate this question, we stained for ciliary marker ARL13b in *PAK2* siRNA cells and counted the number of ARL13b cilia (Fig 8A) and the length (Fig 8B) of the cilium. In *PAK2* knockdown brain ECs, the percent cilia and the length of cilia are significantly smaller compared with control siRNA knockdown brain ECs (Fig 8A and B). To evaluate the contribution of HSPGs to cilium length and numbers, we treated *PAK2* and control siRNA cells with heparinase II and stained for ARL13b, PDGF-BB, and HSPGs. Upon heparinase treatment, control siRNA cells showed little to no difference in the number of cilia or cilium length, indicating that basal HS expression is not associated with cilium morphology or ciliogenesis (Fig 8A and B). Upon *PAK2* siRNA and heparinase II treatment, the number of cilia is not different

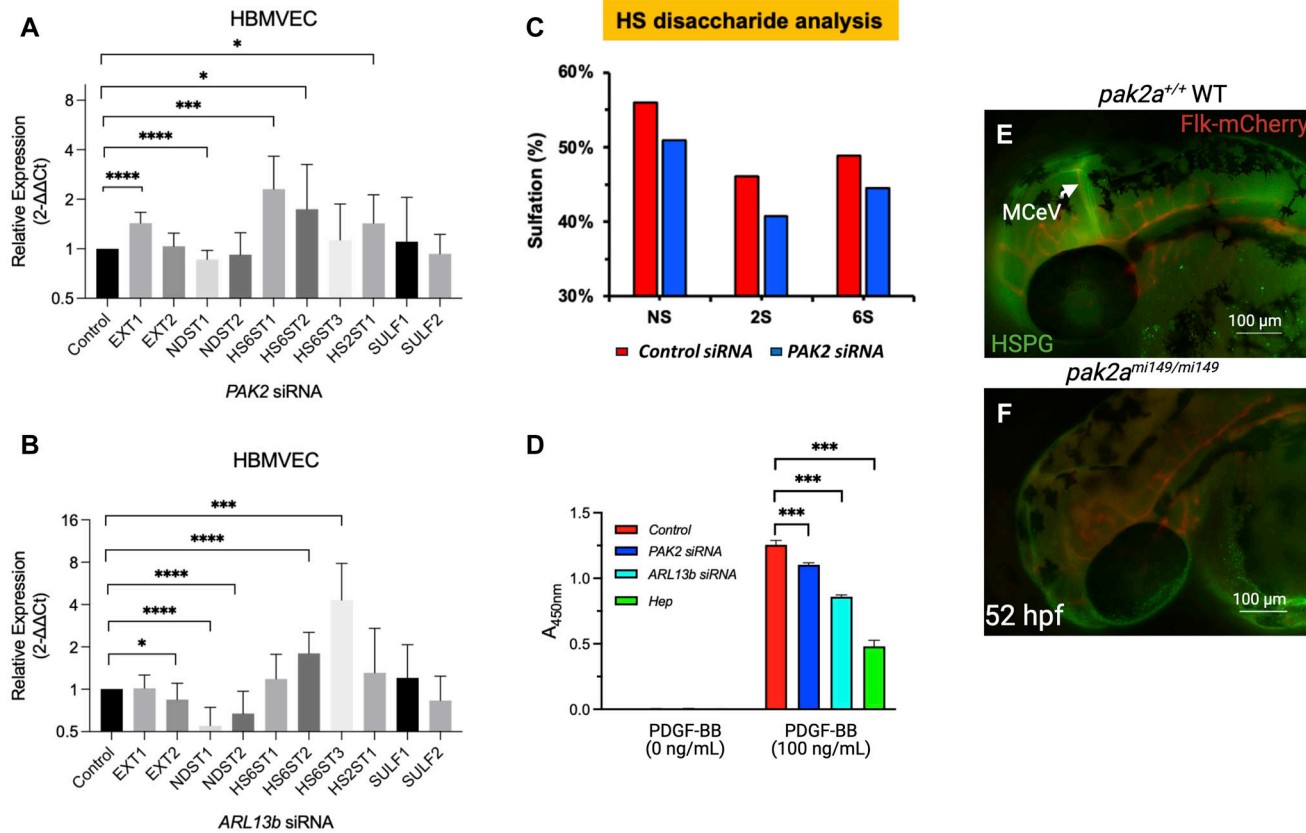

**Figure 7. PAK2 and ARL13b influence PDGF-BB cell surface binding and heparan sulfate modifications in vasculature.**
**(A, B)** qRT-PCR for indicated HS proteoglycan gene synthesis and modifying enzyme targets in *PAK2* siRNA ECs (A) and *ARL13b* siRNA ECs (B), respectively. **(C)** HS disaccharide analysis in control and *PAK2* siRNA brain ECs. **(D)** PDGF-BB cell surface ELISA binding assay on control, *PAK2* siRNA brain ECs, and *ARL13b* siRNA brain ECs. Heparinase treatment is used as a positive control. **(E, F)** Panels are immunofluorescent HSPG antibody stained (green) in *pak2a* WT (E) or *pak2a* homozygous bleeder (F) 52-hpf embryo in a Tg(*flk*:mCherry) (red) fish background. MCeV stands for mid-cerebral vein. The arrow points to region where HSPGs are stained in MCeV, which is surrounding the blood vessel outline. Scale bars in (E, F) are 100 μm. **(A, B)** *T* test was used for statistical analysis of qRT-PCR data in (A, B).

when compared to *PAK2* siRNA-alone cells, suggesting that most of the cilium effect is from *PAK2*-level modulation. However, cilium length in *PAK2* siRNA ECs + heparinase is significantly (*P* = 0.01) smaller than *PAK2* siRNA ECs without heparinase (Fig 8B). We also performed comparable experiments in *ARL13b* siRNA cells and found that cilium numbers were greatly reduced under the *ARL13b* knockdown condition and the effect of heparinase was negligible (Fig 8B). As per cilium length, we found that *ARL13b* siRNA ECs + heparinase have significantly longer cilia (Fig 8B, *P* = 0.032) compared with *ARL13b* siRNA ECs without heparinase. Thus, the heparinase treatment causes opposite effects on cilium length in *PAK2* and *ARL13b* knockdown ECs, which requires additional investigation.

To assess for in vivo EC cilia under *pak2a* modulation, we investigated EC cilium numbers (ARL13b-GFP) along 100 μm of MtA and MCeV vessel length in *pak2a* hom bleeders and compared them with *pak2a* WT. We found that *pak2a* hom bleeders have fewer EC cilia in the MtA and MCeV vessels compared with *pak2a* WT (Fig 8C). We already showed in previous panels that *pak2a* hom bleeders have no HSPGs on the same brain vessels (Fig 7H). The PAK2:EC cilium datasets when combined with data from the previous section (PAK2:HSPG) collectively suggest that PAK2 controls

HSPG deposition and modifications on the cell surface, which partly influences cilium development and PDGF-BB binding on the cell surface.

### PDGF-BB and cilia rescue *pak2a* cerebral hemorrhage phenotype

Because *pak2a* knockdown shows reduced PDGF-BB protein expression in brain ECs and influences HSPG synthesis and sulfation modifications on the abluminal EC surface that are important for PDGF-BB binding, we hypothesized that injecting PDGF-BB protein will rescue cerebral hemorrhage observed in *pak2a* hom bleeders. We injected 1 ng of the human PDGF-BB protein into a clutch of *pak2a* het embryos and imaged embryos that showed the bleeding phenotype compared with bleeders from a vehicle-injected clutch of embryos (Fig 8D and E). Comparing the size and extent of hemorrhage, we observed a decrease in both parameters in PDGF-BB–injected embryos (Fig 8D) compared with vehicle-injected control *pak2a* hom bleeders (Fig 8E). The bleeding area was smaller in PDGF-BB–injected embryos compared with control vehicle–injected embryos (Fig 8F, *P* < 0.01). We also investigated cilium numbers in the PDGF-BB–injected embryos (Fig 8G and G', white arrows) and

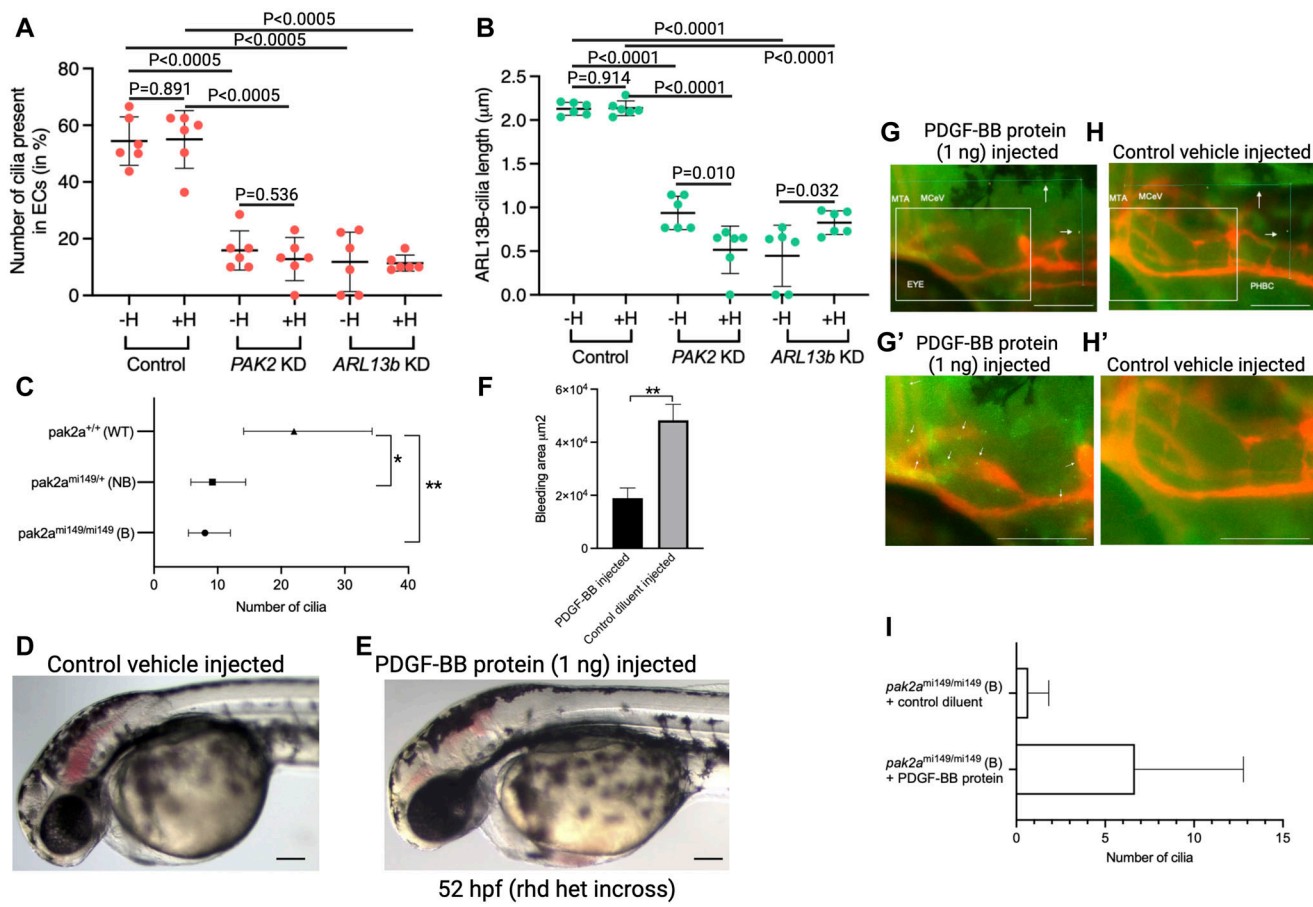

**Figure 8. PAK2:EC cilia:HSPG:PDGF–BB connection in vascular stability.**
**(A, B)** Quantification of EC ARL13b cilium numbers (A) and ARL13b cilium length (B) in control, *PAK2* siRNA brain ECs, and *ARL13b* siRNA brain ECs with and without heparinase (H) treatment. **(C)** Panel shows quantification of the number of EC ARL13b cilia in n = 8 *pak2a* homozygous bleeders (B) (mi149/mi149), n = 6 *pak2a* heterozygous non-bleeders (NB) (mi149/WT), and n = 5 *pak2a* WT (WT/WT). **(D, E)** Whole-mount images of control vehicle (D) or human PDGF-BB protein (1 ng)–injected zebrafish 52-hpf embryo. **(D, E)** Note brain bleeding is reduced in panel (E) compared with panel (D). **(D, E, F)** Quantification of the bleeding area in panels (D, E). N = 6 for control vehicle–injected and N = 5 for PDGFB-BB protein–injected embryos. Scale bars in (D) are 200 *μ*m. **(C)** GLM with negative binomial distribution was performed to compare the number of cilia among group in (C). Dunnett's test was used to adjust for multiple comparisons. Estimated means and 95% confidence intervals (CIs) were used for plotting the number of cilia. WT had more cilia compared with group B or NB, adjusted *P* = 0.0018 and 0.013, respectively. **(F)** Unpaired *t* test was used to analyze bleeding data in (F). **P = 0.0031. **(G, H)** Immunofluorescent ARL13b antibody–stained (green) images of human PDGF-BB protein (1 ng)–injected (G) or control vehicle–injected (H) Tg(*flk*:mCherry) *rhd^mi149/mi149* (homozygous) bleeders. White arrows in (G, H) show Keyence microscope measurement of dimensions of the area analyzed for quantitation of endothelial cilia: 160 *μ*m posterior to the MTA/MCeV and 90 *μ*m dorsal to PHBC. MTA, metencephalic artery; MCeV, middle cerebral vein; PHBC, primordial hindbrain channels. **(G, G′, H, H′)** White boxes in panels (G, H) are magnified in panels (G′, H′). **(G′, H′)** Note: Many cilia are seen in panel (G′) (white arrows) compared with panel (H′). Panel (I) is the quantification of the number of endothelial cilia in the human PDGF-BB protein (1 ng)–injected and control vehicle–injected *rhd^mi149/mi149* (homozygous) bleeders. N = 3 embryos were quantified in each group. Scale bars in (G, G′, H, H′) are 50 *μ*m. An unpaired *t* test was used to analyze differences in the number of cilia, which were found to be not statistically significant.

compared them with control vehicle–injected embryos (Fig 8H and H′). We observed an increase in EC cilium numbers in PDGF-BB–injected embryos compared with control vehicle–injected embryos (Fig 8I). Previously, we already showed that injecting ciliary *ARL13b* mRNA in embryos from incross of the *pak2a* het clutch also rescued the hemorrhage phenotype (Thirugnanam et al, 2022). Taken together, results from our previous work and current work suggest that PDGF-BB can induce PAK2 and ARL13b ciliogenesis signal intracellularly in brain ECs. Ciliogenesis PAK2/ARL13b signals inside the cell can induce PDGF-BB protein expression that is secreted from ECs and found on the basolateral side of developing blood vessels. On the basolateral side, PDGF-

BB binds to HSPGs to create gradients that facilitate pericyte recruitment to promote and maintain embryonic vascular stability (Graphical abstract).

## Discussion

In this study, we report the identification of the PDGF-BB gradient on the basolateral side of brain ECs, which we hypothesize associates with endothelial cilia and HSPGs on the basolateral side to facilitate migration of *pdgfrb*⁺ pericytes closer to the

endothelium to promote vascular stability. Salient features of this study include the identification of brain endothelial cilia on the basolateral side, identification of PDGF-BB gradients in vivo on the basolateral side in close apposition to brain ECs, the identification of HSPGs in the PDGF-BB-PAK2 and ciliogenesis mechanism that is critical for vascular stability, and the identification of PDGF-BB as a potential vascular stabilizing factor in vivo.

To date, endothelial cilia have been widely considered as blood flow sensors (Hierck et al, 2008; Nauli et al, 2008) and thus primarily considered to be located on the apical side of ECs. The presence of endothelial cilia on the basolateral side has been observed previously in lymphatic vessels (Paulson et al, 2021). In brain collateral arteries, basolateral ciliary pockets have been identified (Zhang et al, 2019), but whether these ciliary pockets emerged from basolateral or apical cilia is not fully clear. Here, we show the presence of brain endothelial cilium basolateral location in cortex vessels in zebrafish, mice, and humans (Fig 3B–D). Quantification of basolateral brain endothelial cilium location in mice and humans shows that this is the preferred site on these vessels compared with apical location (Figs 3E and S2). In addition, brain microvascular ECs subjected to flow conditions in vitro in a microfluidic device also show the predominant basolateral location, thus collectively arguing for a critical function for the basolateral endothelial cilia. However, the functional role of basolateral endothelial cilia in embryonic brain vascular development is not known. Our studies highlight the potential for basolateral brain endothelial cilia to serve as contributors of ligands such as PDGF-BB to extracellular space to perform paracrine and juxtacrine function such as recruitment of supporting mural cells to promote vascular stability (Graphical abstract). Support for this idea comes from initial observation wherein we identified that brain endothelial cilia contain PDGF-BB (Fig 6A and B). We present evidence here that brain ECs secrete PDGF-BB in culture but whether PDGF-BB is secreted from brain cilium–associated extracellular vesicles (Mohieldin et al, 2021) or non-ciliary extracellular vesicles is not known. We also showed here that PAK2 and ciliogenesis signal ARL13B induce the release of PDGF-BB from brain ECs. Thus, multiple signals control PDGF-BB release from brain ECs, which can act on brain ECs in an autocrine manner or on mural cells in a paracrine/juxtacrine manner. For <u>autocrine</u> effects, we previously showed that PDGF-BB–like VEGF-A can promote EC ciliogenesis in brain ECs through VEGF receptor-2 (Graphical abstract) (Thirugnanam et al, 2022). In this study, we show that PAK2 and ciliary ARL13b signaling promotes secretion of PDGF-BB from brain ECs (Fig 4A and B), which promotes <u>paracrine/juxtacrine</u> PDGFR-$\beta$–expressing mural cell recruitment (see Graphical abstract).

What happens to PDGF-BB once released from brain ECs? Evidence in the literature shows that specific heparin-binding sequences in PDGF-BB promote its interaction with HSPGs on the endothelial basolateral surface (Lindblom et al, 2003; Abramsson et al, 2007). This is a key point as mural cells also express HSPGs, but it is the endothelial HSPGs in the central nervous system that facilitate PDGF-BB–mediated PDGFR-$\beta$ activation (Stenzel et al, 2009). Inductive reasoning suggests that PDGF-BB secreted from brain ECs interacts with HSPGs juxtaposed to the endothelium and stays proximal to brain ECs to form a gradient or concentration depots of the PDGF-BB ligand. This idea was suggested by Lindblom et al (2003) and Betsholtz (2004), but to our knowledge, the PDGF-BB gradient or concentration depots were not observed in vivo. Here, using PDGF-BB–specific antibodies, PDGF-BB ligand gradients were observed juxtaposed to mouse cortex brain ECs and zebrafish brain ECs (Fig 4C–G). Furthermore, the PDGF-BB gradient was disturbed in pak2a mutant fish and in Ext1 EC KO mice (Fig 5), which collectively argues for an endothelial-specific PAK2a-mediated PDGF-BB gradient formation surrounding the brain vasculature.

The expression of enzymes of the HSPG synthesis and modification was significantly altered in PAK2 siRNA and ARL13b siRNA ECs, and global sulfation was affected on the brain EC surface of PAK2 siRNA cells, which directly influenced the binding of PDGF-BB to the EC surface (Fig 6). N-sulfation in HSPGs is known to be critical for PDGF-BB binding (Abramsson et al, 2007), which supports our findings. Down-regulation of Ndst1 in PAK2 siRNA ECs is expected to reduce cell surface N-sulfation of HSPGs, which is known to be important for PDGF-BB binding (Abramsson et al, 2007). HSPG protein staining around zebrafish brain vessels in pak2a mutant is less, which directly supports the fewer PDGF-BB gradient formation data around pak2a mutant brain vessels (Fig 6E and F).

Finally, the clinical implication of this finding is directly relevant to intraventricular hemorrhage, a pediatric condition, where during embryonic development, dysregulation of vascular development processes is considered as a major contributing factor to the etiology of unstable brain vessels in the germinal matrix leading to hemorrhage (Ballabh, 2014). We injected human PDGF-BB protein in a 1-cell-stage embryo derived from cross of pak2a heterozygous adults, which rescued the brain bleeding phenotype in zebrafish (Fig 8E and F). This remarkable finding is consistent with the interpretation that diminishment of the PDGF-BB protein in pak2a rhd vessels is a causal event to vascular instability. We also observed increased EC cilia in PDGF-BB–injected embryos (Fig 7I). Cilium-dense vessels include angiogenic central arteries (CtAs). In comparison, cilium-poor vessels include MCeV and PHBC. Interestingly, the PDGF-BB gradient is not proximal to cilium-rich vessels and pericytes with shorter arm are often found next to CtAs with cilia. These results collectively argue for pericyte preference to cilium-rich vessels to promote vascular stabilization. Previously, injecting human ARL13b mRNA in embryos from mating of pak2a heterozygous adults also rescued the bleeding phenotype, and this rescue was dependent on when (24 versus 30 hpf) the mRNA injection occurred (Thirugnanam et al, 2022). As both Pak2 and Arl13b promote PDGF-BB secretion from brain ECs, these data collectively argue that PDGF-BB is a key molecule when up-regulated at the right embryonic time (24 hpf) facilitates embryonic brain vascular stability. However, how the PDGF-BB gradient facilitates pericyte recruitment to cilium-dense vessels remains uncertain. Limitations of our study include the lack of functional test in mammals for PAK2–cilium–PDGF-BB interactions, specifically the endothelial cell–autonomous PAK2 contribution and the rescue of the signaling pathway in mammals.

A summary model illustrating the proposed PAK2–cilium–PDGF-BB–HSPG signaling axis and its role in embryonic vascular stability is provided in the Graphical abstract. For PDGF-BB to facilitate embryonic brain vascular stability temporally, spatial interaction with HSPGs on basolateral EC surface is necessary to promote a

gradient of PDGF-BB ligand formation, which will help recruit *pdgfrb*⁺ mural cells to the ECs. Our findings here imply that the PDGF-BB autocrine endothelial signaling loop promotes endothelial ciliogenesis and PDGF-BB secretion, which creates a gradient of the PDGF-BB ligand surrounding the vasculature in association with HSPG. The PDGF-BB:HSPG association in a juxtacrine fashion promotes *pdgfrb*⁺ mural cell recruitment to facilitate vascular stability during vertebrate embryonic development.

# Materials and Methods

### Zebrafish

All studies on zebrafish were carried out under approved protocol AUA320 at the Medical College of Wisconsin (MCW). Zebrafish were kept in tanks on a recirculating aquatic housing system. The RT and water temperature were maintained at 28.5°C. The triple-transgenic zebrafish line Tg(*pdgfrb*: GFP; *kdrl*: mCherry; and *beta-actin*: Arl13b-GFP) that labels star-shaped pericytes (green), blood vessels (red), and string-shaped cilia (green), respectively, was generated by crossing three single-transgenic lines. The Tg(*beta-actin*: Arl13b-GFP) line was obtained from Dr. Brian Ciruna, Toronto, Canada. The Tg(*pdgfrb*:Gal4;UAS-GFP) line (Ando et al, 2016) was obtained from Dr. Naoki Mochizuki, Osaka, Japan, and Tg(*kdrl*: mCherry) lines were obtained from ZIRC. Although pericytes and cilia are both labeled with GFP, they have distinct shapes (Fig 2A'–C'). The zebrafish *redhead* (*rhd^{mi149}*) (Buchner et al, 2007) mutant line used in this study was obtained from Dr. Sarah Childs, Toronto, Canada. A small percentage of the *rhd^{mi149}* homozygous embryos develop brain hemorrhage between 48 and 50 h post-fertilization (hpf). The *rhd^{mi149}* has a point mutation (SNP) (T/A) in the *pak2a* gene. For more information about the hypomorphic mutation and the percentage of embryos that develop brain hemorrhage, please refer to Buchner et al (2007). In this study, homozygous *rhd^{mi149}* mutants have been designated as *pak2a^{mi149/mi149}*, heterozygous *rhd^{mi149}* mutants as *pak2a^{mi149/+}*, and WT fish as *pak2a^{+/+}*. The WT fish in Figs 1 and 4 are *pak2a^{+/+}* genotypes, which were generated from an incross of *pak2a^{mi149/+}* heterozygous adults. Heterozygous mutant adults (*pak2a^{mi149/+}*) were crossed into fluorescent triple-transgenic fish described above. Fluorescent embryos were raised to adulthood and genotyped to identify fluorescent fish that were heterozygous for the *rhd* mutation (*pak2a^{mi149/+}*). In Figs 2, 5, and 7, the WT fish is triple-transgenic in the AB background. Details for genotyping protocol are available in supplemental materials. For Fig 3, WT controls were a standard control MO injected (2 ng) into triple-transgenics in the AB background. All embryos included in quantification of phenotypes were genotyped.

### Genotyping *rhd^{mi149}* fish adults and embryos

A custom SNP genotyping assay request for the *rhd^{mi149}* mutation was submitted online using the Thermo Fisher Scientific custom assay design tool. SNP mutation details were as follows: ZDB-GENE-021011-2, Chr 2: 36628880 (GRCz11), ZDB-ALT-071121-

1, mi149/redhead, point mutation (T/A). A TaqMan SNP genotyping assay for the *rhd^{mi149}* mutation was designed by Thermo Fisher Scientific (Assay ID: ANZTY9A, Cat# 4332077). Please refer to the protocol online (https://www.thermofisher.com/document-connect/document-connect.html?url=https://assets.thermofisher.com/TFS-Assets%2FLSG%2Fmanuals%2FMAN0009593_TaqManSNP_UG.pdf) for further details on the assay.

### Zebrafish embryo mounting and 3D imaging

Embryos were dechorionated at the desired stage (52 hpf) and were anesthetized in 0.016% Tricaine and then mounted in a 1% solution of low melting agarose in fish water in a glass-bottom dish. The low-melt agarose was cooled to 37°C before adding to the embryo for mounting. Post-solidification of the agarose, the embryos were imaged on a Keyence microscope at RT. The objectives for imaging in all figures except Fig 7 were performed using CFI Plan Apo 40x, Numerical Aperture 0.95, WD 0.21 mm, water lens. We used CFI Plan Apo 20x, Numerical Aperture 0.75, WD 1.0 mm for Fig 7. An area of the zebrafish embryo brain 190 µm posterior to the mid-cerebral vein (MCeV) and 150 µm dorsal to the primordial hindbrain channels (PHBC) was imaged using a Keyence microscope and analyzed for pericytes and cilia (Fig 2). For Fig 3D, the number of cilia (green) was counted in 150 µm of MtA and 150 µm of MCeV starting at the base of these vessels where they intersect the PHBC. For Fig 8G–I, the number of endothelial cilia was quantified in a region of the head 160 µm posterior to the MTA/MCeV and 90 µm dorsal to PHBC. For z-stack imaging, an appropriate pitch was selected to capture ~34 continuous images. The first slice of the z-stack was chosen starting at the depth of focus (upper limit) at which the blood vessel metencephalic artery (MtA), MCeV, central arteries (CTAs), and PHBC in the brain area first came into focus. The end position (lower limit) of the z-stack was chosen at the depth of focus where the vessels ended just before they went out of focus. For more details on Keyence z-stack imaging, refer to our previous publication (Thirugnanam et al, 2022).

### Zebrafish injections

Zebrafish protein injections were carried out with human recombinant PDGF-BB protein (PHG0045; Life Technologies) (1 ng/embryo), which was injected into entire clutches of one-cell-stage embryo of a cross between heterozygous mutant adults (*pak2a^{mi149/+}*). An equivalent volume of vehicle was injected into separate clutches at the one-cell stage as control for the experiment.

### Mice

Mouse studies in MCW were performed under approved protocols AUA1022 and AUA7184 and at USF were performed under AUA IS00011193. The ARL13b-EGFP transgenic mice in the C57BL/B6 background were procured from Clapham laboratory at HHMI Janelia Research Campus (Delling et al, 2013). The conditional *Ext1* flox (*Ext1^{fl/fl}*) mice (Inatani et al, 2003; Kraushaar et al, 2010; Qiu et al, 2013, 2018) were bred with *Tie2Cre* transgenic mice (Kisanuki et al, 2001; Wang et al, 2005; Zhang et al, 2014) to generate *Tie2Cre⁺Ext1^{fl/fl}* (*Ext1^{ECKO}*) mice. For Fig 5G–L, *Tie2Cre⁺Ext1^{fl/fl}*

(*Ext1*<sup>ECKO</sup>) embryos were compared with littermate WT *Ext1*<sup>fl/fl</sup> controls (*Ext1*<sup>fl/fl</sup>).

## Mouse qRT-PCR primers

Primers used include the following: *Ext1* (Forward: 5′-GCTCTTGTC TCGCCCTTTTGT-3′, Reverse: 5′-GTGGTGCAAGCCATTCCTAC-3′), *Ext2* (Forward: 5′-ATGTGTGCGTCGGTCAAGTAT-3′, Reverse: 5′-AGAATGGGG CCAAAACTGAAA-3′), *Ndst1* (Forward: 5′-CTGCCTGTTCAGCGTTTTCAT-3′, Reverse: 5′-CGAGTAGAGGCTCTCCACAAA-3′), *Ndst2* (Forward: 5′-TTCCTGGCTTATTATGTGTCCAC-3′, Reverse: 5′-GGGTTCAGTTCGAGCTGT CT-3′), *Hs2st1* (Forward: 5′-GCTCCTCAGGATTATGATGCC-3′, Reverse: 5′-TTTCTCGGACTTCGTGTCTTG-3′), *Hs6st1* (Forward: 5′-ACGCCCAGG AAGTTCTACTAC-3′, Reverse: 5′-GTTGTACGGGCAGTCCATGAA-3′), *Hs6st2* (Forward: 5′-CCCCGAAAGGCGTCTTCTTC-3′, Reverse: 5′-CAT GGGGCCGAAAATCTTGGA-3′), *Hs6st3* (Forward: 5′-TCCAGTGTCACGTTA CCTGAG-3′, Reverse: 5′-TGTAGGTGCAATCCATAAACTCC-3′), *Sulf1* (Forward: 5′-GATCCCCGAGGTTCAGAGGA-3′, Reverse: 5′-GGTGTAGTC ACAAAGGCATTGA-3′), *Sulf2* (Forward: 5′-GGCAGGTTTCAGAGGGACC-3′, Reverse: 5′-GAAGGCGTTGATGAAGTGCG-3′), *Pak2* (Forward: 5′-CGACTCCAACACAGTGAAGCAG-3′, Reverse: 5′-TCACTACTGCGGGTGCTT CTGT-3′), *Arl13b* (Forward: 5′-GAACCAGTGGTCTGGCTGAGTT-3′, Reverse: 5′-GTTTCAGGTGGCAGCCATCACT-3′), and *Gapdh* (reference gene) (Forward: 5′-GTCTCCTCTGACTTCAACAGCG-3′, Reverse: 5′-ACC ACCCTGTTGCTGTAGCCAA-3′). All primers were synthesized by Integrated DNA Technologies (IDT).

## Human samples

The sampling and handling of fetal human specimens conformed to the ethical rules of the Section of Molecular Pathology, Department of Precision and Regenerative Medicine and Ionian Area, University of Bari, and approval was gained from the local Ethics Committee of the National Health System in compliance with the principles stated in the Declaration of Helsinki. Fetal tissue was collected after obtaining informed consent from the mother at the end of the abortion procedure.

## Cell culture

Primary human brain microvascular endothelial cells (HBMECs) (Cat# ACBRI 376; Cell Systems Corporation) were maintained at 37°C in a 5% $CO_2$ incubator in an endothelial cell complete medium (Cat# C22010; PromoCell). All cell culture wells were seeded equally, and wells were randomized to control versus experimental conditions with duplicates or triplicates per condition. All experiments were performed between passages 4 and 6. Respective siRNAs from Horizon-inspired cell solutions *Control* (Cat# D-001810-01-05), *ARL13B* (Cat# J-017365-09-0005), *PDGF-BB* (Cat# J-011749-05-0005), and *PAK2* (Cat# J-003597-10-0005) at the concentration of 25 nM were transfected using Lipofectamine 2000 reagent (Cat# 11668019; Gibco) and incubated for 48 h. Heparinase (heparinase I/III blend from Flavobacterium heparinum (Cat# H3917-50UN; Sigma-Aldrich)) concentration used in the experiments was 0.2 IU/ ml for a time period of 2 h.

## Primary brain EC transfection and Western blotting

Briefly, HBMECs were seeded into six-well culture dishes for Western blotting ~24 h before transfection. Respective siRNA (control and PDGF-BB) was transfected using Lipofectamine 2000 reagent (Cat# 11668019; Gibco) and incubated for 48 h. Cell lysates were collected from HBMECs using RIPA buffer (Cat# R2078; Sigma-Aldrich) supplemented with cOmplete Mini EDTA-free protease inhibitor cocktail (Cat# 11836170001; Roche) and PhosSTOP phosphatase inhibitor (Cat# 4906845001; Roche). Protein concentration in clarified lysates was determined using a bicinchoninic acid (BCA) protein assay (Thermo Fisher Scientific) according to the manufacturer's instructions, and equal amounts of total protein (20–30 µg per lane) were mixed with 4× Laemmli sample buffer, boiled for 5 min at 95°C, and resolved by SDS–PAGE. Proteins were transferred to PVDF membranes, blocked in 5% nonfat dry milk in TBST, and probed with the following primary antibodies: ARL13B (Cat# 17711-1-AP; Proteintech), PAK2 (Cat# 2608S; Cell Signaling Technology), PDGF-BB (Cat# NBP1-58279; Novus Biologicals), and β-actin (Cat# MA1-744; Thermo Fisher Scientific) as a loading control. HRP-conjugated anti-rabbit (Cat# 7074; Cell Signaling Technology) and anti-mouse (Cat# 7076; Cell Signaling Technology) secondary antibodies were used for chemiluminescence detection. Band intensities were quantified by densitometry using ImageJ software (NIH). For each blot, background signal was subtracted, and the intensity of each target band was normalized to the corresponding β-actin band from the same lane. Normalized values were then expressed as a fold change relative to control siRNA-treated cells. Data represent the mean ± SEM from at least three independent biological experiments.

## 3D microfluidic assay

To generate 3D EC vessels, we used the Nortis ParVivo triple-channel microfluidic chips, which we will refer to as the microphysiological systems (MPS). To form each chamber's extracellular matrix, collagen type I (7 mg/ml) (Corning Life Sciences) was injected into microfluidic chips via injection ports into collagen chambers followed by incubation at 37°C overnight for polymerization. Upon polymerization, glass mandrels that run through the middle of each chamber are extracted to form a tubular scaffold with a diameter of 125 µm. HBMECs were cultured in EC growth medium in T-75 flasks. Once cells reached confluency, they were detached using TrypLE and suspended in solution at a dilution of three million cells/ml. Cells are then seeded (2.5 µl) into the MPS via cell injection ports, and flow is blocked to trap cells within the surrounding matrix for 45 min before opening perfusion valves to release cells that have not adhered to the matrix. The MPS are placed in incubators within their perfusion modules and are subjected to continuous perfusion of medium at the flow rate of 1 µl/min (shear stress of 0.9 dyne/cm$^2$). Fully confluent EC tubes form over the course of 7–10 d in the MPS.

## 3D MPS staining and imaging

The following process was performed by perfusing solutions lumenally at 6 µl/min using silicone microtubing connected to syringes containing solutions on a syringe pump. EC tubes were

fixed using 4% PFA for 20 min followed by a PBS wash for 20 min. ECs were then permeabilized with 0.2% Triton solution for 1 h at RT, followed by a PBS wash for 20 min. Blocking buffer (PBS, 2% BSA, 5% NGS, 0.1% NaN$_3$) was perfused (20 min) and trapped in the MPS for overnight incubation at 4°C. After incubation, EC tubes were washed with PBS for 20 min, followed by perfusion of the primary antibody solution containing ARL13B (Cat# 17711-1-AP; Proteintech) and acetylated $\alpha$-tubulin (Cat# 32-2700; Thermo Fisher Scientific) both at a dilution of 1:200 (20 min), followed by a 48-h incubation at 4°C. EC tubes were then washed with PBS for 20 min, and the previous step was repeated with secondary antibody (Cat# ab150088; ab150077; Abcam) solution (1:300) containing DAPI (1:300) for nuclear staining. ECs were then washed with PBS for 20 min again before imaging. Fixed and stained EC tubes were then imaged with a Leica SP8X confocal system (Leica DMI6000 microscope) using a 25X water immersion objective.

## qRT-PCR assays

### Endothelial cells

Total cellular RNA was extracted using the TRIzol reagent (Invitrogen) following the manufacturer's protocol, and cDNA was synthesized using the iScript cDNA synthesis kit (Cat#1708891; Bio-Rad) according to the manufacturer's instructions. Quantitative RT–PCR was performed in 20 $\mu$l reaction volumes containing 10 $\mu$l of iTaq Universal SYBR Green Supermix (Cat#1725124; Bio-Rad), 5 $\mu$l of diluted RT product as a template, and 20 pmol of specific primers. The primer sequences for all genes are provided in the supplemental material document. PCR amplification was performed for 40 cycles at 95°C for 5 s and 60°C for 30 s using the Bio-Rad CFX96 real-time PCR system. For each target gene, average Ct values from six replicates were normalized to GAPDH Ct values, and relative gene expression was calculated using the $2^{-\Delta\Delta Ct}$ method.

## ELISA

Briefly, HBMECs were grown in a six-well dish, and the cells were transfected with control, ARL13B, and PAK2 siRNA with and without heparinase; 100 $\mu$l of the cell-free supernatant was used for the assessment of PDGF-BB. The protocol was followed according to the manufacturer's instructions (Human PDGF-BB ELISA Kit, Cat# ELH-PDGFBB-1; RayBiotech).

## Human brain microvascular endothelial cell surface PDGF-BB binding assay

The primary HBMEC surface PDGF-BB binding assay was conducted as previously described (Wang et al, 2005; Qiu et al, 2018; Zhao et al, 2020; Yue et al, 2021; Mah et al, 2023; McMillan et al, 2024b). Cells were seeded at a density of 4 × 10$^4$ cells per well in 96-well plates. After overnight culture were washed and fixed in 4% PFA at RT for 15 min. After washing and blocking in 2% BSA overnight at 4°C, human PDGF-BB-His (Cat# 10572-H07Y; Sino Biological) at a concentration of 1,000 ng/ml in the 2% BSA blocking solution was applied and incubated for 90 min at RT. Wells with PDGF-BB at 1,000 ng/ml plus heparin at 100 $\mu$g/ml were included as HS-dependent controls. Cells were then washed and incubated with

anti-His antibody conjugated with HRP (Cat# 652504; BioLegend) at a dilution of 1:4,000 in 2% BSA for 1 h at RT. After extensive washing, ELISA substrate was applied. After 20 min, color development was stopped with 0.2 M H$_2$SO$_4$. Absorbance at 450 nm was measured to quantify cell surface PDGF-BB binding, after subtracting the value of background wells, which had no PDGF-BB.

## Heparan sulfate structural analysis

HBMECs transfected with PAK2 siRNA or control siRNA were cultured in six-well plates, rinsed with PBS, and lysed with 1 ml of 0.1 M NaOH, followed by the addition of 8 $\mu$l of acetic acid (Wang et al, 2002, 2005, 2023; Qiu et al, 2018). The cell lysate was treated with actinase E (Kaken Biochemicals) (5 mg/ml) and digested overnight at 55°C. After deactivating actinase E at 100°C for 30 min, the digested solution was transferred to a 3,000 MW cutoff filter (Thermo Fisher Scientific). CHAPS (final concentration 2%) was added and mixed with the digested solution in the filter of the spin tube. The filter unit was washed three times with 400 $\mu$l distilled water to enrich released HS. Next, 300 $\mu$l of digestion buffer (50 mM ammonium acetate containing 2 mM calcium chloride, pH 7.0) was added to the obtained HS samples, along with recombinant heparin lyases I, II, and III (expressed in E. coli in Fuming Zhang laboratory) (Lohse & Linhardt, 1992), and digested at 37°C overnight. The resultant HS disaccharides were collected by filtering the digested HS through the 3,000 MW cutoff filter, and the filter unit was washed twice with 200 $\mu$l distilled water. All filtrates containing HS disaccharides were combined and freeze-dried. The dried samples were then labeled with 2-aminoacridone (AMAC) (Sigma-Aldrich) for LC-MS analysis. A mixture containing all HS disaccharide standards prepared at 1 ng/$\mu$l was similarly AMAC-labeled and used as an external standard. HS disaccharide standards (UA-GlcNAc, UA-GlcNS, UA-GlcNAc6S, UA2S-GlcNAc, UA2S-GlcNS, UA-GlcNS6S, UA2S-GlcNAc6S, UA2S-GlcNS6S, where UA is 4-deoxy-$\alpha$-L-threo-hex-4-enopyranosyluronic acid) were purchased from Iduron. The AMAC-labeled disaccharides were separated by reversed-phase high-performance liquid chromatography (RP-HPLC) coupled to a triple quadrupole mass spectrometry system equipped with an ESI source (Thermo Fisher Scientific). LC was performed on an Agilent 1200 LC system at 45°C using an Agilent Poroshell 120 ECC18 (2.7 $\mu$m, 3.0 × 50 mm) column. Mobile phase A was 50 mM ammonium acetate aqueous solution, and mobile phase B was methanol. The mobile phase passed through the column at a flow rate of 300 $\mu$l/min with the following gradient: 0–10 min, 5–45% B; 10–10.2 min, 45–100% B; 10.2–14 min, 100% B; 14–22 min, 100–5% B. Injection volume was 5 $\mu$l. The online MS analysis was performed in multiple reaction monitoring (MRM) mode with the following MS parameters: negative ionization mode with a spray voltage of 3,000 V, a vaporizer temperature of 400°C, and a capillary temperature of 250°C.

## Staining protocols

### Fish

Triple-transgenic heterozygous mutant adults (pak2a$^{mi149/+}$) were crossed to each other. At 52 hpf, embryos with bleeding in the brain (bleeders) were separated from the non-bleeders. Both bleeder

and non-bleeder embryos that were single-transgenic (red) (Tg(*kdrl*: mCherry)) were selected. Embryos were dechorionated, anesthetized with Tricaine, and fixed with 4% PFA in PBS for 2 h at 4°C. Embryos were washed twice with PBS and stored in 10% methanol at −20°C overnight. The next day, embryos were washed three times with PBST (PBS containing 0.1% Tween-20) and permeabilized in acetone at −20°C for 8 min. Embryos were then washed 3 times in PBST and blocked in 10% normal goat serum in PBS for 1 h at RT. After this step, embryos were incubated with either PDGF-BB antibody (Cat# sc-365805, 1:50; PDGF-B Antibody; Santa Cruz Biotechnology, Inc.) or HS antibody (Cat# 370255-S, 1:100; Amsbio LLC) at 4°C overnight. The next day, embryos were washed three times, 5 min each with PBST. Embryos were then incubated with Alexa Fluor 488 goat anti-mouse IgG (H+L) cross-adsorbed secondary antibody (green) (Cat# A-11001, 1:250; Thermo Fisher Scientific) in the dark for 2 h. Embryos were then washed with PBST four times, mounted in 50% glycerol/PBS, and imaged using a Keyence microscope. After imaging, all embryos were collected in separate PCR tubes for genomic DNA extraction and genotyping.

For claudin 5b immunofluorescence experiments, non-transgenic heterozygous mutant adults (*pak2a*$^{mi149/+}$) were crossed to each other. Bleeders and non-bleeder embryos were separated. Double immunofluorescence was performed using the same protocol as stated before. Embryos were incubated with two primary antibodies: Claudin 5 antibody (Cat# 4C3C2, 1:50; Thermo Fisher Scientific) and Kdrl antibody (Cat# ES1003, 1:500; Kerafast). Secondary antibodies used were Alexa Fluor 488 goat anti-mouse IgG (H+L) cross-adsorbed secondary antibody (green) (Cat# A-11001, 1:250; Thermo Fisher Scientific) and Alexa Fluor 568 goat anti-rabbit IgG (H+L) cross-adsorbed secondary antibody (red) (Cat# A-11011, 1:250; Thermo Fisher Scientific). Embryos were genotyped according to the protocol in supplemental materials.

### Immunofluorescence staining of mouse embryos

Embryos were isolated at E11.5, immediately frozen in OCT medium (Cat# 23-730-571; Thermo Fisher Scientific), and stored at −80°C. Tail biopsies were taken for genotyping by PCR. Embryos were sectioned at 10 $\mu$m using a cryostat microtome at −20°C, mounted on charged glass slides, and air-dried at RT for 30 min. Samples were fixed in cold acetone for 10 min, then washed three times with 0.1% Tween-20 in PBS (PBST) for 10 min each. Sections were blocked in PBS containing 10% goat serum, 3% BSA, and 0.2% Triton X-100 for 2 h. They were then incubated overnight at 4°C with primary antibodies: rat anti-endomucin IgG1 (sc-5394; Santa Cruz Biotechnology) and rabbit anti-PDGF-B polyclonal IgG (sc-7878; Santa Cruz Biotechnology). Subsequently, sections were incubated with Alexa Fluor 488–conjugated goat anti-rat IgG (A-11008; Thermo Fisher Scientific) and Alexa Fluor 594–conjugated goat anti-rabbit IgG (A-11012; Thermo Fisher Scientific) for 2 h at RT. Slides were mounted with antifade mounting medium. Immunostained tissue images were captured using a confocal laser-scanning microscope (LSM Meta 510). All samples were visually inspected under the microscope, and representative images were obtained at a resolution of 1,024 × 1,024 using a 63× objective lens with 3× optical zoom.

### Cilium immunofluorescence labeling

Transgenic *Arl13b-EGFP* mice (*C57BL/6* background) were anesthetized in an induction chamber using 5% inhaled isoflurane and then moved to the surgical table and positioned on a sterile drape over a heating pad to maintain body temperature. Isoflurane anesthesia was maintained at 1.5% and continuously delivered via a nosecone. To perfuse and fix the brains for immunohistochemical labeling, a thoracotomy was performed to access the heart for transcardiac gravity-fed perfusion to flush and clear circulating blood from the organs with heparinized PBS followed by 4% formaldehyde solution. Brains were placed in a 4% formaldehyde solution overnight before sectioning or paraffin embedding. For immunohistochemical labeling, tissues from ARL13b-EGFP transgenic mice were dehydrated through graded ethanol, cleared with xylene, and paraffin-infiltrated using automated tissue processing (Sakura Tissue TEK, VIP5, and VIP6). All paraffin-embedded tissues were sectioned at 4 $\mu$m and placed on poly-L-lysine–coated slides. Sections were stained immunohistochemically with an automated immunostaining platform Leica Bond RX using standard labeled streptavidin–biotin detection. Briefly, sections were deparaffinized, rehydrated, and protein-blocked (30 min, X0909; DAKO). Antibodies for GFP (Cat# ab290; Abcam) were incubated for 1 h at RT. After TBST washing, biotinylated anti-rabbit secondary antibodies (Cat# 715-066-152; Jackson ImmunoResearch) were incubated for 45 min at RT before streptavidin-HRP and DAB for visualization. Hematoxylin was applied as a nuclear counterstain. Omission of primary antibody was performed as a negative reagent control. High-resolution images were obtained with a Hamamatsu NanoZoomer 2.0-HT high-resolution digital slide scanner.

### Human fetal brain histology and immunostaining

Human fetal brains were from the archive of the Department of Translational Biomedicine and Neuroscience—DiBraiN, Human Anatomy and Histology Unit, Bari University School of Medicine. The samples included in the study were from two fetuses at 22 wk of gestation, spontaneously aborted because of preterm rupture of the placental membranes with no history of neurological pathologies. The fetal age was estimated based on the crown-rump length and/or pregnancy records (counting from the last menstrual period). At autopsy, the fetuses did not reveal macroscopic structural abnormalities and/or malformations of the CNS. The dorsolateral wall of the telencephalic vesicles (future cerebral hemispheres) was dissected along the coronal planes in samples 0.5 cm thick, fixed by immersion in 2% PFA plus 0.2% glutaraldehyde in PBS, pH 7.6, overnight at 4°C, then archived and stored in 0.02% PFA in PBS at 4°C. For the present study, fetal brain samples (n = 6/fetus) were cut using a vibrating microtome (Leica Microsystem) into 30-$\mu$m-thick sections, parts of which were processed for routine histological analysis with toluidine blue staining to rule out the presence of microscopic malformations. All the other sections were stored at 4°C in 0.02% PFA in PBS and submitted to immunofluorescence confocal microscopy protocols. Double immunostainings were carried out with rabbit pAb anti-ARL13B (Cat# 17711-1-AP; Proteintech) at a dilution of 1:300 in blocking buffer (BB; 1% BSA and 2% FCS in PBS) and mouse mAb anti-CD31

(Cat# M0823; Dako) at a dilution of 1:40 in BB. Free-floating sections were washed with 3x PBS before permeabilization with 0.5% Triton X-100 (Cat# 1610407; Bio-Rad) in PBS for 30 min at RT. This was followed by blocking in BB for 30 min at RT and overnight incubation with the combined primary antibodies, at 4°C. Sections were washed with 3x PBS and incubated with the secondary Abs, Alexa Fluor 555 goat anti-mouse (Cat# A21425; Thermo Fisher Scientific) (1:400 dilution in BB) and biotinylated goat anti-rabbit (Cat# BA-100; Vector) (1:400 dilution in BB) for 1 h at RT, the latter then revealed by streptavidin-conjugated Alexa Fluor 488 (Cat# S-32354; Thermo Fisher Scientific) (1:400 dilution in BB) incubated 1 h at RT. The sections were then washed 3x with PBS, and nuclear counterstaining was performed with TO-PRO-3 (Cat# T3605; Thermo Fisher Scientific) (diluted 1:3 K in PBS) for 10 min at RT. Finally, the sections were allowed to adhere to polylysine slides (Menzel-Glaser) by drying for 10 min at RT, coverslipped with Vectashield (Cat# H-1900; Vector Laboratories), and sealed with nail varnish. Negative controls were prepared by omitting the primary antibodies and by mismatching the secondary antibodies. Quantification of ARL13b$^+$ cilia in human CD31$^+$ vessels is described in the supplemental methods document.

### Human fetal brain cilium quantification

Fetal brain telencephalon from 22 wk of gestation was used in the cilium analysis. Quantitative assessment of ARL13B$^+$ cilium density was carried out on three sections per brain by computer-aided morphometric analysis using Leica Confocal Multicolor Package (Leica Microsystems) and ImageJ (NIH, Bethesda, MD) software. The number of ARL13B$^+$ cilia was interactively counted and normalized to the same cumulative vessel length (CVL = multiple of 100 $\mu$m of vessel length). The CD31$^+$ microvessels included in the morphometric analysis ranged from capillaries (9.9 $\mu$m in diameter) to small arterioles and venules (10–30 $\mu$m in diameter). The internal minor diameter and length of CD31$^+$ vessels were automatically calculated using ImageJ measure tools. The numbers of ARL13B$^+$ cilia were categorized based on their precise locations as follows: basolateral (abluminal) or apical (luminal) cilia in the three distinct zones of the telencephalon wall, ventricular zone (VZ), subventricular zone (SVZ), and cortical plate (CP). The accurate identification of ARL13B$^+$ cilia in each of the described categories, abluminal and luminal, primarily depended on the examination of single optical planes of confocal z-stacks at intervals of 0.35 $\mu$m.

### Laser confocal microscopy analysis

The stained sections were examined using a Leica TCS SP8 confocal laser-scanning microscope (Leica Microsystems) with sequential scanning (McMillan et al, 2024a). Confocal images were acquired at 250-nm intervals along the z-axis of the sections using 40x and 63x oil-immersion objectives, with zoom factors ranging from 1.5 to 3. Z-stacks (projection images), and single z-slices were analyzed by Leica confocal software (Multicolor Package; Leica Microsystems).

### HBMEC cilium staining in vitro

Immunofluorescence analysis was performed as described previously (Lauring et al, 2019; Thirugnanam et al, 2022). Briefly, HBMECs

were seeded into six-well plates on coverslips. All the IF experiments for ARL13B and PDGF-BB colocalization and ciliary quantification under *ARL13B* and *PAK2* knockdown with and without heparinase were performed by seeding cells on the same day with similar seeding density and cell synchronization. Cells were washed thrice with 1X PBS (Cat# 10010023; Gibco) and fixed with 4% PFA (Cat# 15710; Electron Microscopy Sciences) for 15 min. Fixed cells were washed again with 1X PBS before permeabilization with 0.1% Triton X-100 (Cat# 1610407; Bio-Rad) for 15 min. This was followed by blocking in 4% BSA in PBS and overnight incubation with primary antibodies of ARL13B (Cat# 17711-1-AP, dilution 1:500; Proteintech) and PDGF-BB (Cat# NBP1-58279, dilution 1:500; Novus Biologicals). Cells were again washed with 1X PBS and incubated with Alexa Fluor 488 anti-rabbit (Cat#A21206; Invitrogen) (1:500) and Alexa Fluor 568 anti-rabbit (Cat# A10042; Invitrogen), for 90 min at RT, and washed before mounting with DAPI (Cat# LS-J1033-10; LifeSpan Biosciences) and imaged using a Zeiss confocal microscope at a magnification of 63X. The number of cilia and their length were quantified by the number of nuclei to cilia using ACDC v0.93 cilium-specific software, which runs on a MATLAB platform as described previously (Lauring et al, 2019).

### Quantification and statistical analysis

#### Pericyte and cilium analysis

Triple fluorescent heterozygous mutant adults ($pak2a^{mi149/+}$) were crossed, and the embryos were collected for imaging and analysis. At 52 hpf, embryos from clutches were divided into two groups: bleeders (B) and non-bleeders (NB). Triple fluorescent embryos were selected, and their hindbrain areas were imaged using a Keyence microscope. Cilium and pericyte numbers and their distances from vessels were measured in the hindbrain region. A rectangular area (length = 190 $\mu$m posterior to the MCeV and width = 150 $\mu$m dorsal to the PHBC) was analyzed in all images (Fig 2). The BZ-X Analyzer software (Keyence) was used for distance and area measurements. The "X-Y measure" tool was used for measuring distances, the "area measure" tool was used for measuring areas, and the "count" tool was used for counting cilia and pericytes. For endothelial cilium quantification (Fig 3D), four control images from double-transgenic zebrafish Tg(*beta-actin*: Arl13b-GFP; *kdrl*: mCherry) were used to quantify abluminal versus luminal endothelial cilia in 52-hpf zebrafish embryo brains. For each image (Fig 3D), the number of cilia (green) was counted in 150 $\mu$m of MtA and 150 $\mu$m of MCeV starting at the base of these vessels where they intersect the PHBC. For PDGF-BB:cilium rescue experiments (Fig 8G–I), we performed immunofluorescence experiment using ARL13B primary antibody (Cat# 17711-1-AP, dilution 1:50; Proteintech) and Alexa Fluor 488 donkey anti-rabbit IgG (H+L) secondary antibody (green) (Cat# A-21206,1:250; Thermo Fisher Scientific). The number of endothelial cilia was quantified in a region of the head 160 $\mu$m posterior to the MTA/MCeV and 90 $\mu$m dorsal to PHBC. All bleeders were genotyped to confirm homozygous ($pak2a^{mi149/mi149}$) genotype. All statistical tests performed on the data are indicated in the figure legends. Data were presented as the mean and standard error of the mean or 95% confidence intervals (CIs) or median and interquartile ranges. The Shapiro–Wilk test was performed to check normality of the data. A two-sample *t* test or Welch's *t* test was used to compare the

differences in outcome measures between two groups. Analysis of variance (ANOVA) with Dunnett's test to adjust for multiple comparisons was used for comparisons among more than two groups. Some data were log-transformed to meet parametric assumptions. The Mann–Whitney–Wilcoxon or Kruskal–Wallis test with Dwass–Steel–Critchlow–Fligner method for multiple comparisons adjustment was employed where parametric assumptions were not satisfied. A chi-squared test or Fisher's exact test was used to examine the associations between the longest arm categories and the genotypes. False discovery rate (FDR) was used to adjust for multiple comparisons. Count data were analyzed by generalized linear model (GLM) with the Poisson or negative binomial distribution. $P < 0.05$ was considered statistically significant. SAS version 9.4 (SAS Institute Inc.) was used for statistical analysis.

### Claudin junction analysis

We quantified the number of claudin junctions in the claudin double immunofluorescence images. In these images, claudin junctions are labeled green and blood vessels are labeled red (Fig 1A–C and A'–C'). The junctions were counted along the PHBC, starting at the base of the MCeV and 80 µm posterior to it (Fig 1A–C and A'–C'). Any point where two or more lines intersect was considered as a junction.

### PDGF-BB gradient quantification
**Zebrafish**
**Number of PDGF-BB sources in head, PHBC, and heart**

A PDGF-BB–positive source was determined as a strong punctate-like staining (Fig 5A and A', white box) with some stained dots showing "Wi-Fi"–like symbol–shaped gradients (Fig 4F'). Sources of PDGF-BB (with and without Wi-Fi rings) were counted in the head, heart, and below PHBC regions. In addition, for the head, the *number of rings within a PDGF-BB gradient* was counted as follows: PDGF-BB–labeled source have "Wi-Fi"–shaped gradient with rings that radiate from the center of the source (Fig 5E and F, green illustration). The number of rings in the gradients emerging from each PDGF-BB–labeled source in the head was counted. For some sources that contain multiple gradients around the center of the source, the average number of rings in all gradients was used as the number of rings for that source. We counted the number of rings in all sources in the head, and average number of PDGF-BB gradient rings for that head was determined. The data plotted (Fig 5E) are the average number of PDGF-BB gradient rings for all heads in the group.

### Distance of the farthest ring from the center of the PDGF-BB source

We also measured the distance of the farthest ring from the center of each source. For sources that have multiple Wi-Fi gradients, the average of the farthest ring distances was used. After determining the farthest ring distance for all the sources in the head, the average farthest ring distance for that head was plotted (Fig 5F).

### Mouse

Radial fluorescence intensity analysis of PDGF-BB and endomucin signals was performed using the *Fiji* Radial Profile Extended plugin that quantified spatial gradients around endothelial cells (Fig 5I'–L').

## Data Availability

The original contributions presented in the study are included in the article; further inquiries can be directed to the corresponding author.

## Supplementary Information

## Acknowledgements

We thank members of the Developmental Vascular Biology Program and collaborators who provided valuable input for this study. S Prabhudesai, K Thirugnanam, KR Rarick, and R Ramchandran were funded through R33HL154254. S Prabhudesai, K Thirugnanam, L Wang, and R Ramchandran are also funded through R01HL179583 grant. Additional support for the Developmental Vascular Biology Program (RR laboratory) came from Department of Pediatrics, Children's Research Institute, Medical College of Wisconsin. L Wang was supported by 5R01HL093339, 4R01AG074289, and 4R01AG069039. The authors of the Department of Translational Biomedicine and Neuroscience, University of Bari, thank Antonella Bizzoca for technical assistance. All figures were created using BioRender.

### Author Contributions

S Prabhudesai: data curation, formal analysis, validation, investigation, visualization, methodology, and writing—review and editing.
K Thirugnanam: data curation, formal analysis, validation, investigation, visualization, methodology, and writing—review and editing.
X Song: validation, investigation, visualization, and methodology.
H Yang: validation, investigation, visualization, and methodology.
M Errede: data curation, validation, investigation, visualization, and methodology.
F Girolamo: validation, investigation, visualization, and methodology.
T Neumann: resources and methodology.
A Marzullo: validation, visualization, and methodology.
S Bafti: resources, software, methodology, and writing—review and editing.
K Vanderhoef: investigation, visualization, and methodology.
KR Rarick: data curation, investigation, visualization, methodology, and writing—review and editing.
AD Spearman: methodology and writing—review and editing.
AY Pan: data curation, formal analysis, validation, and methodology.
CA Alvarez: data curation, investigation, and visualization.
J Yang: data curation, validation, investigation, visualization, and methodology.
F Zhang: data curation, validation, investigation, and visualization.
JS Dordick: software, formal analysis, investigation, and methodology.
D Virgintino: resources, data curation, formal analysis, validation, investigation, visualization, methodology, and writing—original draft.

L Wang: conceptualization, supervision, funding acquisition, validation, investigation, methodology, and writing—review and editing.

R Ramchandran: conceptualization, resources, formal analysis, supervision, funding acquisition, investigation, and writing—original draft, review, and editing.

### Conflict of Interest Statement

R Ramchandran is the President and Founder of a start-up company CIAN, Inc. that is based on developing ciliary protein biomarkers for brain and vascular injury.

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
