## [Reviewer comments · Life Science Alliance]

Brain vascular stability relies on PAK2-cilia-PDGF-BB-HSPGs on basolateral side of endothelium.

Shubhangi Prabhudesai, Karthikeyan Thirugnanam, Xuehong Song, Hua Yang, Mariella Errede, Francesco Girolamo, Thomas Neumann, Andrea Marzullo, Sepand Bafti, Kayla Vanderhoef, Kevin Rarick, Andrew Spearman, Amy Pan, Claudia Alvarez, Jiyuan Yang, Fuming Zhang, Jonathan Dordick, Daniela Virginitino, Lianchun Wang, and Ramani Ramchandran
DOI: <https://doi.org/10.26508/lsa.202503460>

Corresponding author(s): Ramani Ramchandran, Medical College of Wisconsin and Lianchun Wang, University of Georgia

Review Timeline:

Submission Date:	2025-07-21
Editorial Decision:	2025-08-28
Revision Received:	2025-11-24
Editorial Decision:	2025-12-10
Revision Received:	2025-12-17
Accepted:	2025-12-18

Scientific Editor: Sarita Hebbar

Transaction Report:

August 29, 2025

Re: Life Science Alliance manuscript #LSA-2025-03460-T

Ramani Ramchandran
Medical College of Wisconsin
Pediatrics
8701 Watertown Plank Road
Milwaukee, WI 53226

Dear Dr. Ramchandran,

Thank you for submitting your manuscript entitled, "Brain vascular stability occurs through PAK2-endothelial cilia-PDGF-BB-heparan sulfate proteoglycan interface on basolateral side" to Life Science Alliance (LSA). The manuscript was assessed by two expert reviewers, whose comments are appended to this letter.

As you will note, both reviewers found the study of value but have raised some important concerns that preclude publication at this stage. These concerns pertain to presentation of existing data and the need for new experimental data.

--We agree with both reviewers that the quality of all representative images must be substantially improved in terms of clarity to represent the matched descriptions. Further we also agree that quantified data must be accompanied by representative images (Reviewer 1, point 2). Both reviewers have asked for quantification of (1) EC-cilia and (2) gradient levels of PDGF-BB with a suggested approach from Reviewer 2. We concur that these data must be quantified and we leave the approach to your discretion.

--In terms of new experiments, we agree with Reviewer 2 that you must provide some kind of experimental evidence for the temporal sequence of events (for cilia loss and PDGF-BB) with loss of PAK2. We also concur with Reviewer 1 that you must provide information (citing publications or experimental evidence) if basolateral EC-cilia can be observed in other cell types such as HUVECs.

--Finally we encourage you to follow the recommendation from Reviewer 2 that extrapolation to mammalian systems must be discussed with the caveat of missing experimental evidence, or backed with experimental evidence wherever possible.

In line with the overall recommendations, we invite you to submit a revised manuscript. When submitting the revision, please include a letter addressing the reviewers' comments point by point. While a rebuttal must respond to all points in some form, additional data to resolve these points is not required.

Thank you for this interesting contribution to Life Science Alliance. We are looking forward to receiving your revised manuscript.

Sincerely,

Sarita Hebbar, PhD
Scientific Editor
Life Science Alliance
<http://www.lsjournal.org>

B. MANUSCRIPT ORGANIZATION AND FORMATTING:

Reviewer #1 (Comments to the Authors (Required)):

There is evidence that cilia in vascular endothelial cells (ECs) are critical for embryonic brain vascular stability. However, the mechanisms by which EC-cilia function during vascular development remain unknown.

In this study, Prabhudesai et al. demonstrate that brain EC-cilia are predominantly located in the basolateral location. PDGF-BB secreted from brain ECs is distributed in a gradient pattern, which is regulated by p21-activated kinase protein (PAK2) and heparan sulfate proteoglycans (HSPGs).

Overall, the results provided are interesting findings on the mechanisms by which endothelial cilia regulate vascular permeability via the regulation of PDGF-BB-HSPG interaction. However, the authors should improve the manuscript by addressing the following points.

Comments:

1. In Fig. 1, the zoomed images are somewhat unclear to show cell junctions. Better images need to be presented.

2. Fig. 2 only has the quantitative results. Representative images should be provided.

3. It would be interesting to examine whether basolateral EC-cilia formation can be observed in other human ECs such as HUVECs, compared to the brain ECs (Fig. 3a). High magnification images (Fig. 2, b' and b'') seem unclear. It is difficult to appreciate the basolateral EC-cilia formation in Fig. 3c, and high magnification images need to be presented. Similar to Fig. 3d, quantification of EC-cilia can be performed in Fig. 3a-c.

4. Gradient levels of PDGF-BB in the mouse and zebrafish brains seem unclear (Fig. 4). Is there any way to quantitatively measure gradient patterns of PDGF-BB?

5. The authors state that in brain ECs, PDGF-BB is only co-localized to one of two cilia. Why does it happen?

Reviewer #2 (Comments to the Authors (Required)):

The manuscript by Prabhudesai et al. investigates the role of the PAK2-endothelial cilia-PDGF-BB-HSPG axis in maintaining

embryonic brain vascular stability. Using zebrafish, mouse, human fetal brain samples, and cultured human brain microvascular endothelial cells, the authors demonstrate that PAK2 regulates endothelial cilia formation, PDGF-BB production, and heparan sulfate proteoglycan (HSPG) modification, which collectively contribute to pericyte recruitment and vascular stability. The study integrates multiple model systems to propose a novel mechanistic framework. However, several key aspects of the proposed mechanism require further clarification and experimental support. In its current form, the causal relationships within the PAK2-cilia-PDGF-BB-HSPG pathway remain partially unresolved, and additional experimental work is necessary to strengthen the conclusions. Moreover, some figures suffer from insufficient image clarity, which limits the interpretability of the presented data.

Major issues:

1. The authors propose a positive feedback loop whereby PAK2 promotes cilia formation, cilia facilitate PDGF-BB production, and PDGF-BB in turn promotes ciliogenesis. However, the directionality of this relationship remains unclear. It is not demonstrated whether PDGF-BB reduction precedes cilia loss in *pak2a* mutants, or whether cilia loss is the initial defect that subsequently reduces PDGF-BB secretion. The authors need to perform time-course experiments in *pak2a* mutants (and/or PAK2 siRNA cells) to determine the temporal sequence of PDGF-BB downregulation versus cilia loss. In addition, rescue experiments restoring PDGF-BB in the absence of cilia, or vice versa, would help establish the hierarchy of events.
2. The data suggest that PAK2 knockdown alters the expression and sulfation of HS-modifying enzymes, but it is unclear whether this effect is direct or mediated through cilia-dependent signaling changes. The authors need to test whether restoring cilia formation in PAK2-deficient cells rescues HSPG sulfation patterns. Conversely, they need to disrupt HSPG modification without affecting cilia to determine if PDGF-BB binding loss still occurs, thereby separating cilia-related and HSPG-related effects.
3. Much of the causality is inferred from zebrafish and in vitro human EC models, with extrapolation to mammalian systems. Although mouse brain sections were used for PDGF-BB gradient visualization, functional tests of PAK2-cilia-PDGF-BB interactions in a mammalian system are lacking. The authors need to perform targeted disruption of endothelial PAK2 or cilia in mice, combined with PDGF-BB rescue, to strengthen the translational relevance. If experimental conditions are not available, further research suggestions can be discussed in detail.

Minor points:

1. The immunostaining images suggest gradient formation, but the gradient profiles are not quantitatively presented. Plotting fluorescence intensity across defined regions would provide more objective evidence.
2. While pericyte proximity to ciliated vessels is described, it would be useful to quantify pericyte coverage versus cilia density under different conditions, and to assess whether pericytes preferentially associate with cilia-rich segments even when PDGF-BB is absent.
3. Several fluorescence images suffer from high background and low contrast, making it difficult to discern fine structures such as short endothelial cilia.
4. Some figure panels lack clear scale bars or have inconsistent labeling of magnification. This should be standardized.

RESPONSE LETTER TO REVIEWER and EDITOR COMMENTS**Editor comments**

Comment: As you will note, both reviewers found the study of value but have raised some important concerns that preclude publication at this stage. These concerns pertain to presentation of existing data and the need for new experimental data.

Response: We are happy to note that the reviewers found value in our study. To address the remaining concerns, we are including a revised manuscript with track changes (made in response to the reviewer comments) along with a clean copy of the manuscript. A point-by-point response below to each comment is also provided. We have made the following changes in response to the reviewer comments, which includes new data, better images and more mechanistic insight into the signaling pathway.

Figure 1. Better higher magnification images (**1a''-1c''**) showing junctions are provided.

Figure 2. The figure panels (**2a-c'**) are provided that match the quantification in the panels.

Figure 3. Quantification provided for zebrafish (**3c-c'**, **3d**), and mouse (**3e**, **e'**, **e''**) panels.

Figure 5. Quantification provided for PDGF-BB gradients in fish (**5e**, **5f**) and mouse (**5i'**, **5l'**).

Figure 6. New figure with new data to evaluate the autocrine PDGF-BB signaling loop in ECs (**6b**, **6c**).

Figure 7. Added *ARL13b* siRNA evaluation in panels b and d.

Figure 8. Added *ARL13b* siRNA evaluation +/- Heparinase in panels a and b. New data is provided in panels g-i that investigate whether cilia rescue occurs in PDGF-BB injected embryos.

Figure 9. A new figure showing the conceptual and mechanistic model is provided that is based on the data in this manuscript.

Comment: We agree with both reviewers that the quality of all representative images must be substantially improved in terms of clarity to represent the matched descriptions. Further we also agree that quantified data must be accompanied by representative images (Reviewer 1, point 2). Both reviewers have asked for quantification of (1) EC-cilia and (2) gradient levels of PDGF-BB with a suggested approach from Reviewer 2. We concur that these data must be quantified, and we leave the approach to your discretion.

Response: As mentioned above, we have provided enhanced images for Fig.1. For Fig. 2, we have re-made the figure and included the representative images that match the quantification shown in the panels. New data showing quantification of EC-cilia (**Fig. 8c-i**) and gradient levels of PDGF-BB (**Figs. 5e, f, i', l'**) has been provided. Methodology for quantification has been updated in the methods section.

Comment: In terms of new experiments, we agree with Reviewer 2 that you must provide some kind of experimental evidence for the temporal sequence of events (for cilia loss and PDGF-BB) with loss of PAK2.

Response: Temporal sequence of events is best demonstrated *in vivo*. However, we have a phenotype issue that precludes us from performing this experiment *in vivo*. The *pak2a* mutant fish show bleeding phenotype only at 48 hours post fertilization, and thus we are unable to distinguish mutants from wild type or heterozygous embryos any earlier in development to assess cilia distribution or PDGF-BB gradient loss differences at earlier time points. Therefore, temporality of the mechanism is difficult to establish with the genetic mutant.

Our next best option was to investigate *PAK2* siRNA ECs, which reviewer 2 suggested in point 1 below to help establish the hierarchy of events. As *ARL13b* was also involved in this signaling pathway, we added *ARL13b* (cilia) loss-of-function signaling experiments (**Fig. 7b**, **7d**, **Fig. 8a-b**) in the resubmission. *PAK2* siRNA ECs (or *ARL13b* siRNA ECs) show less PDGF-BB (**Fig. 4a-b**) and cilia numbers (new data - **Fig. 8a-b**). Previously, in brain ECs, we showed that PDGF-BB induced *PAK2* and *ARL13b* levels and cilia numbers in ECs.¹ Both datasets together suggests that PDGF-BB-*PAK2*/*ARL13b*-*CILIA*-PDGF-BB autocrine signaling loop exists in ECs (**Fig. 9b**). The predictive outcome here would be that if we knocked down the ligand PDGF-BB, we should observe less *PAK2*/*ARL13b* and cilia in ECs, which we indeed observed (**new data – Fig. 6**). Thus, the autocrine EC signaling loop that we have identified here

is perhaps timed in a manner that it helps maintain PDGF-BB gradient formation in conjunction with heparan sulfate proteoglycan (HSPGs) at cell surface. In this model, EC-cilia could be the source of PDGF-BB (**Fig. 6a-a'**) or could simply assist HSPGs in maintaining the PDGF-BB gradient, which are hypotheses that need testing. We have included this model interpretation (**Fig. 9b**) and limitations of the model in the discussion section.

Comment: We also concur with Reviewer 1 that you must provide information (citing publications or experimental evidence) if basolateral EC-cilia can be observed in other cell types such as HUVECs.

Response: EC-cilia on the basolateral side has been observed in lymphatic ECs.² We have included this citation in the manuscript. Lymphatic ECs do arise from venous ECs during development. As HUVECs are venous EC, we would speculate that HUVECs would carry EC-cilia on basolateral side.

Unfortunately, we are unable to determine EC-cilia location in HUVECs because the microfluidic system at Nortis that was used for brain ECs is no longer in the market.

Comment: Finally, we encourage you to follow the recommendation from Reviewer 2 that extrapolation to mammalian systems must be discussed with the caveat of missing experimental evidence or backed with experimental evidence wherever possible.

Response: We have included the limitations in terms of lack of congruent *in vivo* PAK2 mammalian data sets in the discussion section.

Reviewer #1 (Comments to the Authors (Required):

Overall, the results provided are interesting findings on the mechanisms by which endothelial cilia regulate vascular permeability via the regulation of PDGF-BB-HSPG interaction. However, the authors should improve the manuscript by addressing the following points.

Response: We thank the reviewer for this comment and have strived to improve the manuscript based on the provided recommendations.

1. In Fig. 1, the zoomed images are somewhat unclear to show cell junctions. Better images need to be presented.

Response: We have provided clearer zoomed-in images (**Fig.1a'-c'**) to show the cell junctions. The new zoomed images of the EC-junctions clearly show loss of junctions in primordial hindbrain channel (PHBC) vessel in *pak2a* bleeders compared to heterozygous (dotted EC junctions) or wild type (clear line EC junctions) embryos.

2. Fig. 2 only has the quantitative results. Representative images should be provided.

Response: Figure 2 has been completely redone to include the representative images along with the quantification. The confocal images (**2a-c'**) are provided that match the quantification in the panels.

3. It would be interesting to examine whether basolateral EC-cilia formation can be observed in other human ECs such as HUVECs, compared to the brain ECs (Fig. 3a). High magnification images (Fig. 2, b' and b'') seem unclear. It is difficult to appreciate the basolateral EC-cilia formation in Fig. 3c, and high magnification images need to be presented. Similar to Fig. 3d, quantification of EC-cilia can be performed in Fig. 3a-c.

Response: We have already responded to the HUVEC comment earlier. We quantified the luminal vs. abluminal cilia location in both zebrafish (**3c-c'**, **3d**), and mouse (**3e**, **e'**, **e''**) brains panels. As the data shows, in mammals, the brain endothelial cilia show a proclivity for abluminal location.

4. Gradient levels of PDGF-BB in the mouse and zebrafish brains seem unclear (Fig. 4). Is there any way to quantitatively measure gradient patterns of PDGF-BB?

Response: We have quantified both the mouse and zebrafish brain PDGF-BB gradients, which are now included in figure 5. The PDGF-BB gradients in fish (**5e**, **5f**) brain were compared between *pak2* WT and *pak2* bleeders. In mouse brain, the PDGF-BB gradients (**5i'**, **5l'**) were compared between *Ext1* WT and *Ext1* knockout. The details associated with this quantification is included in the methods section.

5. The authors state that in brain ECs, PDGF-BB is only co-localized to one of two cilia. Why does it happen?

Response: This is an interesting point, for which we can only offer speculative explanations. Our previous publication showed that brain ECs have occasionally two cilia,³ and the second cilium arises from daughter centriole. The primary cilium arises from the mother centriole. We are not sure if PDGF-BB is in the mother centriole-derived cilium or daughter centriole-derived cilium. Depending on the cilium origin, we speculate that the functions of PDGF-BB could vary. It is noteworthy that treating brain ECs with PDGF-BB does increase the ciliary length of both cilia.³ Thus, the PDGF-BB presence in cilium is a topic that needs more investigation.

Reviewer #2 (Comments to the Authors (Required)):

The manuscript by Prabhudesai et al. investigates the role of the PAK2-endothelial cilia-PDGF-BB-HSPG axis in maintaining embryonic brain vascular stability. Using zebrafish, mouse, human fetal brain samples, and cultured human brain microvascular endothelial cells, the authors demonstrate that PAK2 regulates endothelial cilia formation, PDGF-BB production, and heparan sulfate proteoglycan (HSPG) modification, which collectively contribute to pericyte recruitment and vascular stability. The study integrates multiple model systems to propose a novel mechanistic framework.

Response: We thank the reviewer for appreciating the integration of multiple model systems to provide a novel mechanistic framework for embryonic vascular stability in vertebrates.

Major issues:

1. The authors propose a positive feedback loop whereby PAK2 promotes cilia formation, cilia facilitate PDGF-BB production, and PDGF-BB in turn promotes ciliogenesis. However, the directionality of this relationship remains unclear. It is not demonstrated whether PDGF-BB reduction precedes cilia loss in *pak2a* mutants, or whether cilia loss is the initial defect that subsequently reduces PDGF-BB secretion. The authors need to perform time-course experiments in *pak2a* mutants (and/or *PAK2* siRNA cells) to determine the temporal sequence of PDGF-BB downregulation versus cilia loss. In addition, rescue experiments restoring PDGF-BB in the absence of cilia, or vice versa, would help establish the hierarchy of events.

Response: We have provided a detailed response to this comment in the first section of the response letter. Essentially, we do not believe that there is a hierarchy of events here, instead it is a continuum of autocrine signaling in ECs that prepares the ECs for pericyte recruitment and vascular stability. The role of abluminal EC-cilia in this process is that it is a likely source of PDGF-BB for proximal gradient formation, which occurs in association with HSPGs on the abluminal side. Other functions for EC-cilia in this process cannot be excluded. We interpret here that EC-cilia help HSPGs form the PDGF-BB gradient, and act as a guide for future pericyte migration. In terms of rescue, we have indeed performed this experiment and showed that PDGF-BB injection in *pak2a rhd* bleeders rescues bleeding (**Fig. 8e**) and cilia numbers (new **Fig. 8g¹-i**). We have also included a detailed conceptual (**Fig. 9a**) and mechanistic hypothesis (**Fig. 9b**) in new **figure 9** that provides clarity to the findings from this study.

2. The data suggest that PAK2 knockdown alters the expression and sulfation of HS-modifying enzymes, but it is unclear whether this effect is direct or mediated through cilia-dependent signaling changes. The authors need to test whether restoring cilia formation in PAK2-deficient cells rescues HSPG sulfation patterns. Conversely, they need to disrupt HSPG modification without affecting cilia to determine if PDGF-BB binding loss still occurs, thereby separating cilia-related and HSPG-related effects.

Response: To directly address the ciliary involvement in HSPGs, we downregulated ARL13b ciliogenesis signal in brain ECs (new data- **Fig. 6b**). Indeed, qPCR changes showed robust alteration to HSPG enzyme expression (new data- **Fig. 7b**). We also determined PDGF-BB binding on *ARL13b* siRNA cells, which was altered (new data- **Fig. 7d**). *HS6ST2*, *HS6ST3* and *NDST1* enzymes are regulated in both *PAK2* and *ARL13b* siRNA ECs compared to control ECs. Further, we also assessed cilia length and numbers in the *PAK2* and *ARL13b* siRNA ECs with and without heparinase (new data- **Fig. 8a-b**). No changes in cilia numbers are noticed with and without heparinase in either gene knockdown condition. However, a distinct change in cilia length is noticed, where in upon heparinase

treatment, *ARL13b* siRNA ECs + heparinase have longer cilia compared to *ARL13b* siRNA ECs without heparinase. This effect was not observed in *PAK2* siRNA ECs +/- heparinase. We have a molecular explanation for this distinct effect, wherein under *ARL13b* siRNA ECs + heparinase condition, the *PAK2* levels are upregulated (**Fig. R1**), which in turn would induce PDGF-BB and in turn cilia length. In this interpretation, heparinase and *ARL13b* (cilia signal) combination attenuates intracellular signaling networks (PAK2-PDGF-BB) to induce ciliary length. This data will be part of subsequent comprehensive work that investigate PAK2-ALR13b signaling nexus in ECs in the context of PDGF-BB secretion and HSPG binding. We provide this data here for the reviewer. Thus, collectively, disrupting a cilia signal (*ARL13b* siRNA) directly induces HSPG alterations and PDGF-BB binding to EC surface, and disrupting HSPGs (heparinase treatment) on the cell surface also influences intracellular signals (PAK2) that induce ciliogenesis (*ARL13b*-PDGF-BB). Separating HSPG from cilia signaling in ECs is thus challenging as each is regulated by the other.

3. Much of the causality is inferred from zebrafish and *in vitro* human EC models, with extrapolation to mammalian systems. Although mouse brain sections were used for PDGF-BB gradient visualization, functional tests of PAK2-cilia-PDGF-BB interactions in a mammalian system are lacking. The authors need to

[Figure removed by editorial staff per authors' request]

perform targeted disruption of endothelial PAK2 or cilia in mice, combined with PDGF-BB rescue, to strengthen the translational relevance. If experimental conditions are not available, further research suggestions can be discussed in detail.

Response: We acknowledge the limitations of our approach and attempted to provide EC relevance by directly modulating the targets under study in brain ECs. The *Pak2* flox mice have recently been procured via MTA and yet to be transferred to our collaborator's Prof. Lianchun Wang's facility in University of South Florida. Our future studies will assess the endothelial deletion of *Pak2* and ciliogenesis in brain microvascular stability as part of a recently funded R01 grant from NIH. We have included the limitations in the mammalian studies associated with this manuscript in the discussion section.

Minor points:

1. The immunostaining images suggest gradient formation, but the gradient profiles are not quantitatively presented. Plotting fluorescence intensity across defined regions would provide more objective evidence.

Response: We have now provided quantitative evidence for the PDGF-BB gradients (**Fig. 5**). The PDGF-BB gradients in fish brain (**5e, 5f**) and the PDGF-BB gradients in the mouse brains (**5i', 5l'**) were quantified.

2. While pericyte proximity to ciliated vessels is described, it would be useful to quantify pericyte coverage versus cilia density under different conditions, and to assess whether pericytes preferentially associate with cilia-rich segments even when PDGF-BB is absent.

Response: This is an interesting and insightful point, which we explored in our current data. To check for cilia-dense vs. cilia-poor vascular segments, we investigated *pak2a* WT fish brains at 52 hpf because *pak2a* bleeders show fewer PDGF-BB gradients (**Fig. 5d-e**) and contain very few cilia overall. As shown

(**Fig. R2a**), we found that EC-cilia were enriched in emerging angiogenic central arteries (CtAs) compared to MCeV or PHBC – established vessels at 52 hpf. The PDGF-BB gradient also seems to be excluded from the CtA region (**Fig. R2b**). This data therefore suggests that PDGF-BB gradient does not seem to be proximal to cilia-rich vascular segments in 52 hpf brain. As majority of pericytes in *pak2a* WT brain have short arms (**Fig. S1**), and found next to CtAs (**Fig. R2a**), taken together, pericytes do preferentially associate with cilia-dense vessels in the absence of PDGF-BB. We thank the reviewer for this intelligent point. We have brought this point in the discussion section of the manuscript.

3. Several fluorescence images suffer from high background and low contrast, making it difficult to discern fine

structures such as short endothelial cilia.

Response: We have carefully evaluated the fluorescent images to ensure that the main point of our observations is coming through in the figure panels.

4. Some figure panels lack clear scale bars or have inconsistent labeling of magnification. This should be standardized.

Response: We have labeled the scale bars to reflect the magnification in all panels.

[Figure removed by editorial staff per authors' request]

References:

1. Thirugnanam K, Prabhudesai S, Van Why E, Pan A, Gupta A, Foreman K, Zennadi R, Rarick KR, Nauli SM, Palecek SP, et al. Ciliogenesis mechanisms mediated by PAK2-ARL13B signaling in brain endothelial cells is responsible for vascular stability. *Biochem Pharmacol.* 2022;202:115143. doi: 10.1016/j.bcp.2022.115143
2. Paulson D, Harms R, Ward C, Latterell M, Pazour GJ, Fink DM. Loss of Primary Cilia Protein IFT20 Dysregulates Lymphatic Vessel Patterning in Development and Inflammation. *Front Cell Dev Biol.* 2021;9:672625. doi: 10.3389/fcell.2021.672625
3. Thirugnanam K, Gupta A, Nunez F, Prabhudesai S, Pan AY, Nauli SM, Ramchandran R. Brain microvascular endothelial cells possess a second cilium that arises from the daughter centriole. *Front Mol Biosci.* 2023;10:1250016. doi: 10.3389/fmolb.2023.1250016

December 10, 2025

RE: Life Science Alliance Manuscript #LSA-2025-03460-TR

Prof. Ramani Ramchandran
Medical College of Wisconsin
Pediatrics
8701 Watertown Plank Road
Milwaukee, WI 53226

Dear Dr. Ramchandran,

Thank you for submitting your revised manuscript entitled "Brain vascular stability relies on PAK2-cilia-PDGF-BB-HSPGs on basolateral side of endothelium.". Your revised manuscript was evaluated by the reviewers of the original submission; their comments are appended below. You will note that they both have commented that your revised manuscript has addressed all their previous concerns.

We would be happy to publish your paper in Life Science Alliance pending final revisions necessary to meet our formatting guidelines.

-In the legend for Figure 3e, please specify the genotype of mouse brains described. Also include it in the methods. Likewise please specify nature of the controls used in Fig. 5g,h,i.

-We encourage you to incorporate the supplemental methods into the methods section of the main manuscript.

-We request you to complete the description in the methods section:

1. Include complete details on various imaging methods specifically for temperature of imaging (live imaging) and details of objectives used (NA and name, for all imaging)
2. Complete the details in Western blotting section on method for protein quantification.
3. Provide primers used for genotyping of fish
4. For all animal experiments, please specify details for controls ('WT') lines.

-You have the option of using Figure 9 as a Graphical Abstract in which case you would have to remove this figure file and upload it instead as a graphical abstract file, and edit the text accordingly.

-Please add ORCID ID for secondary corresponding author.

-Please add the X and Bluesky handles of your host institute/organisation, as well as your own and/or one of the authors, in our system.

-Please consult our manuscript preparation guidelines <https://www.life-science-alliance.org/manuscript-prep> and make sure your manuscript sections are in the correct order.

-LSA does not permit citation of "data not shown," "manuscript in preparation," "manuscript submitted," etc., in any section of the manuscript. For example: provide evidence to back the statement for sentence starting in line 48 about severe vascular defects in the mouse mutant line, that currently refers to a ms in preparation. If you cannot provide the evidence to back the claim, please remove it.

-Please clearly label the "Conflict of interest" statement in the manuscript file.

-A "Data Availability" section should be placed after the Materials & Methods section. Please consult our guidelines at <https://www.life-science-alliance.org/manuscript-prep#format>.

-Please add your main and supplementary figure legends to the main manuscript text after the references section

-Please be sure that the authorship listing and order is correct.

LSA now encourages authors to provide a 30-60 second video where the study is briefly explained. We will use these videos on social media to promote the published paper and the presenting author (for examples, see <https://docs.google.com/document/d/1-UWCfbE4pGcDdcgzcmiuJl2XMBJnxKYeqRvLLrLSo8s/edit?usp=sharing>). Corresponding or first-authors are welcome to submit the video. Please submit only one video per manuscript. The video can be emailed to contact@life-science-alliance.org

A. FINAL FILES:

B. MANUSCRIPT ORGANIZATION AND FORMATTING:

Thank you for your attention to these final processing requirements. Please revise and format the manuscript and upload materials as soon as you are able.

Sincerely,

Sarita Hebbar, PhD
Scientific Editor
Life Science Alliance
<http://www.lsajournal.org>

Reviewer #1 (Comments to the Authors (Required)):

The authors have sufficiently addressed my comments, and the revised manuscript is significantly improved.

Reviewer #2 (Comments to the Authors (Required)):

The authors have addressed my concerns.

We are providing a point-by-point response below to the final changes requested by the editorial team. Our response to each item is in green font below.

-In the legend for Figure 3e, please specify the genotype of mouse brains described. Also include it in the methods. Likewise please specify nature of the controls used in Fig. 5g,h,i. ✓ Requested changes have been made. See lines 506-510, and lines 1147-1152.

-We encourage you to incorporate the supplemental methods into the methods section of the main manuscript. ✓ We have included the supplemental methods into the methods section of the main manuscript.

-We request you to complete the description in the methods section:

1. Include complete details on various imaging methods specifically for temperature of imaging (live imaging) and details of objectives used (NA and name, for all imaging) – ✓ We have included the requested information in the zebrafish embryo mounting and 3D imaging section in methods (lines 476-483).

2. Complete the details in Western blotting section on method for protein quantification. ✓ Done. (lines 551-571).

3. Provide primers used for genotyping of fish – ✓ We have provided the details for genotyping of fish based on Taqman genotyping SNP assay protocol. We contacted the company for primer and probe details, which are proprietary and confidential. These primers and probes can be ordered using the catalog number and assay ID, which is both provided in the method section (lines 467-474).

4. For all animal experiments, please specify details for controls ('WT') lines. ✓ Updated in figures and in the methods section. Lines 457-465.

-You have the option of using Figure 9 as a Graphical Abstract in which case you would have to remove this figure file and upload it instead as a graphical abstract file, and edit the text accordingly. – We have removed this figure and included it as a graphical abstract file online. A sentence has been added in the last paragraph of the discussion section to reflect this. A summary model illustrating the proposed PAK2–cilia–PDGF-BB–HSPG signaling axis and its role in embryonic vascular stability is provided in the Graphical Abstract. We have also provided a separate document, which contains the legend for the graphical abstract.

-Please add ORCID ID for secondary corresponding author. – Ling Wang's ORCID is included here - 0000-0003-1454-2719. The online LSA system only allows one ORCID link to an article.

Ramchandran's ORCID is included on the LSA system online. We have included both ORCIDs on the first page of the revised manuscript.

-Please add the X and Bluesky handles of your host institute/organisation, as well as your own and/or one of the authors, in our system. We are providing this information here. <https://x.com/MedicalCollege>. LinkedIn handle for Ramchandran - <https://www.linkedin.com/in/ramani-ramchandran-08b084246>

-Please consult our manuscript preparation guidelines <https://www.life-science-alliance.org/manuscript-prep> and make sure your manuscript sections are in the correct order. We have formatted the manuscript as per the LSA guidelines.

-LSA does not permit citation of "data not shown," "manuscript in preparation," "manuscript submitted," etc., in any section of the manuscript. For example: provide evidence to back the statement for sentence starting in line 48 about severe vascular defects in the mouse mutant line, that currently refers to a ms in preparation. If you cannot provide the evidence to back the claim, please remove it. ✓ We have removed the line MS in preparation sentence.

-Please clearly label the "Conflict of interest" statement in the manuscript file. ✓ This has been taken care of.

-A "Data Availability" section should be placed after the Materials & Methods section. Please consult our guidelines at <https://www.life-science-alliance.org/manuscript-prep#format>. ✓ We have done that.

-Please add your main and supplementary figure legends to the main manuscript text after the references section ✓ We have done that.

-Please be sure that the authorship listing and order is correct. ✓ Yes, all correct.

December 18, 2025

RE: Life Science Alliance Manuscript #LSA-2025-03460-TRR

Prof. Ramani Ramchandran
Medical College of Wisconsin
Pediatrics
8701 Watertown Plank Road
Milwaukee, WI 53226

Dear Dr. Ramchandran,

Thank you for submitting your Research Article entitled "Brain vascular stability relies on PAK2-cilia-PDGF-BB-HSPGs on basolateral side of endothelium.". It is a pleasure to let you know that your manuscript is now accepted for publication in Life Science Alliance. Congratulations on this interesting work.

DISTRIBUTION OF MATERIALS:

Again, congratulations on a very nice paper. I hope you found the review process to be constructive and are pleased with how the manuscript was handled editorially. We look forward to future exciting submissions from your lab.

Sincerely,

Sarita Hebbar, PhD
Scientific Editor
Life Science Alliance
<http://www.lsjournal.org>